# Stepwise activation mechanism of the scramblase nhTMEM16 revealed by cryo-EM

Valeria Kalienkova[1], Vanessa Clerico Mosina[2], Laura Bryner[1], Gert T Oostergetel[2], Raimund Dutzler[1]*, Cristina Paulino[2]*

[1]Department of Biochemistry, University of Zurich, Zurich, Switzerland; [2]Department of Structural Biology, Groningen Biomolecular Sciences and Biotechnology Institute, University of Groningen, Groningen, The Netherlands

**Abstract** Scramblases catalyze the movement of lipids between both leaflets of a bilayer. Whereas the X-ray structure of the protein nhTMEM16 has previously revealed the architecture of a $Ca^{2+}$-dependent lipid scramblase, its regulation mechanism has remained elusive. Here, we have used cryo-electron microscopy and functional assays to address this question. $Ca^{2+}$-bound and $Ca^{2+}$-free conformations of nhTMEM16 in detergent and lipid nanodiscs illustrate the interactions with its environment and they reveal the conformational changes underlying its activation. In this process, $Ca^{2+}$ binding induces a stepwise transition of the catalytic subunit cavity, converting a closed cavity that is shielded from the membrane in the absence of ligand, into a polar furrow that becomes accessible to lipid headgroups in the $Ca^{2+}$-bound state. Additionally, our structures demonstrate how nhTMEM16 distorts the membrane at both entrances of the subunit cavity, thereby decreasing the energy barrier for lipid movement.
DOI: https://doi.org/10.7554/eLife.44364.001

*For correspondence:
dutzler@bioc.uzh.ch (RD);
c.paulino@rug.nl (CP)

Competing interests: The authors declare that no competing interests exist.

## Introduction

The movement of lipids between both leaflets of a phospholipid bilayer, which is also referred to as lipid flip-flop, is energetically unfavorable since it requires the transfer of the polar headgroup from its aqueous environment across the hydrophobic core of the membrane (*Kornberg and McConnell, 1971*). Consequently, spontaneous lipid flip-flop is rare and occurs at a time scale of several hours. Whereas the sparsity of this event permits the maintenance of lipid asymmetry, there are processes that require the rapid transbilayer movement of lipids (*Bevers and Williamson, 2016*; *Sanyal and Menon, 2009*). Proteins that facilitate this movement by lowering the associated high energy barrier are termed lipid scramblases (*Pomorski and Menon, 2006*; *Williamson, 2015*). Lipid scramblases are generally non-selective and do not require the input of energy. They participate in various cellular functions ranging from lipid signaling during blood coagulation and apoptosis to the synthesis of membranes, cell division and exocytosis (*Nagata et al., 2016*; *Whitlock and Hartzell, 2017*). First structural insight into lipid scrambling was provided from the X-ray structure of the fungal protein from *Nectria haematococca*, termed nhTMEM16 (*Brunner et al., 2014*). This protein is part of the TMEM16 family (*Milenkovic et al., 2010*), which encompasses calcium-activated ion channels and lipid scramblases with a conserved molecular architecture (*Brunner et al., 2016*; *Caputo et al., 2008*; *Falzone et al., 2018*; *Schroeder et al., 2008*; *Suzuki et al., 2013*; *Suzuki et al., 2010*; *Yang et al., 2008*). As common for the family, nhTMEM16 is a homodimer with subunits containing a cytoplasmic domain and a transmembrane unit composed of 10 membrane-spanning segments. The lipid permeation path is defined by a structural entity termed the 'subunit cavity', which is located at the periphery of each subunit. In the X-ray structure, which was obtained in the presence

of $Ca^{2+}$, this site exposes a continuous polar cavity that spans the entire membrane and that is of appropriate size to harbor a lipid headgroup (*Brunner et al., 2014*). The 'subunit cavity' is believed to provide a pathway for the polar moieties of lipids across the membrane, whereas the apolar acyl chains remain embedded in the hydrophobic core of the bilayer (*Bethel and Grabe, 2016*; *Brunner et al., 2014*; *Jiang et al., 2017*; *Lee et al., 2018*; *Malvezzi et al., 2018*; *Stansfeld et al., 2015*). This process resembles the credit-card mechanism for lipid scrambling, which was previously proposed based on theoretical considerations (*Pomorski and Menon, 2006*). Although in this structure $Ca^{2+}$ ions were identified to bind to a conserved site in the transmembrane domain, the mechanism of how $Ca^{2+}$ activates the protein and how the protein interacts with the surrounding lipids remained elusive. To gain insight into these questions, we have determined structures of the protein in $Ca^{2+}$-bound and $Ca^{2+}$-free conformations by single particle cryo-electron microscopy (cryo-EM), both in detergent and in a membrane environment and we have characterized the effect of mutants on the activation process. Our data reveal two essential features of the protein. They show how the protein deforms the membrane to lower the energy barrier for lipid movement and to steer translocating lipids towards the subunit cavity. Additionally, they define the conformational changes leading to the opening of the subunit cavity, which is shielded from the membrane in a $Ca^{2+}$-free conformation, but exposes a hydrophilic furrow for the permeation of lipids upon $Ca^{2+}$ binding in a stepwise process involving coupled ligand-dependent and ligand-independent steps.

## Results

### Structural characterization of nhTMEM16 in detergent

For our investigations of the ligand-induced activation mechanism of the lipid scramblase nhTMEM16, we have recombinantly expressed the protein in the yeast *S. cerevisiae*, purified it in the detergent n-Dodecyl β-D-maltoside (DDM) in the absence of $Ca^{2+}$ and added $Ca^{2+}$ during sample preparation for structure determination and transport experiments. We have reconstituted the purified protein into liposomes of different composition containing fluorescently labeled lipids and characterized lipid transport using an assay that follows the irreversible decay of the fluorescence upon addition of the membrane-impermeable reducing agent dithionite to the outside of the liposomes (*Brunner et al., 2014*; *Malvezzi et al., 2013*; *Ploier and Menon, 2016*). In these experiments we find a $Ca^{2+}$-dependent increase of transport but also a pronounced basal activity of the protein in absence of $Ca^{2+}$, which was previously described as a hallmark of fungal TMEM16 scramblases (*Brunner et al., 2014*; *Malvezzi et al., 2013*) (*Figure 1—figure supplement 1*). For the structural characterization of nhTMEM16, we have initially studied $Ca^{2+}$-bound and $Ca^{2+}$-free samples in detergent by cryo-EM. By this approach, we have obtained two datasets of high quality and of sufficient resolution (*i.e.* 3.6 Å for $Ca^{2+}$-bound and 3.7 Å for $Ca^{2+}$-free conditions) for a detailed interpretation by an atomic model (*Figure 1*; *Figure 1—figure supplements 2–4*; *Table 1*). In the structure determined in the presence of $Ca^{2+}$, we find a conformation of the protein, which closely resembles the X-ray structure with a root mean square deviation (RMSD) of Cα atoms of 0.66 Å between the two structures (*Figure 1D,E*). The high similarity prevails irrespective of the absence of crystal packing interactions and despite of the fact that the sample used for cryo-EM was only briefly exposed to $Ca^{2+}$ at a comparably low (*i.e.* 300 μM) concentration, whereas for the X-ray structure, the protein was exposed to $Ca^{2+}$ during purification and crystallization at a 10-fold higher concentration (*Brunner et al., 2014*). Cryo-EM density at the conserved $Ca^{2+}$-binding site, contained within each subunit of the homodimer, clearly indicates the presence of two bound $Ca^{2+}$ ions located at the position originally identified in the X-ray structure (*Figure 1F*) (*Brunner et al., 2014*). Remarkably, we found a very similar structure of the protein in the absence of $Ca^{2+}$, thus underlining the stability of this conformation in detergent solution (*Figure 1G,H*). Although both datasets display equivalent conformations of the protein, the structure determined in the absence of $Ca^{2+}$ clearly shows an altered distribution of the divalent ligand in its binding site, concomitant with a weaker density of the C-terminal part of α6 indicating its increased mobility (*Figure 1G,I*; *Figure 1—figure supplement 3*). The same region was previously identified as key structural element in the $Ca^{2+}$-regulation of TMEM16A (*Dang et al., 2017*; *Paulino et al., 2017a*; *Peters et al., 2018*). In the cryo-EM density, both strong peaks of the ligand found in the $Ca^{2+}$-bound conformation are absent and instead replaced by weak residual density at low contour located at the position of the lower $Ca^{2+}$ ion

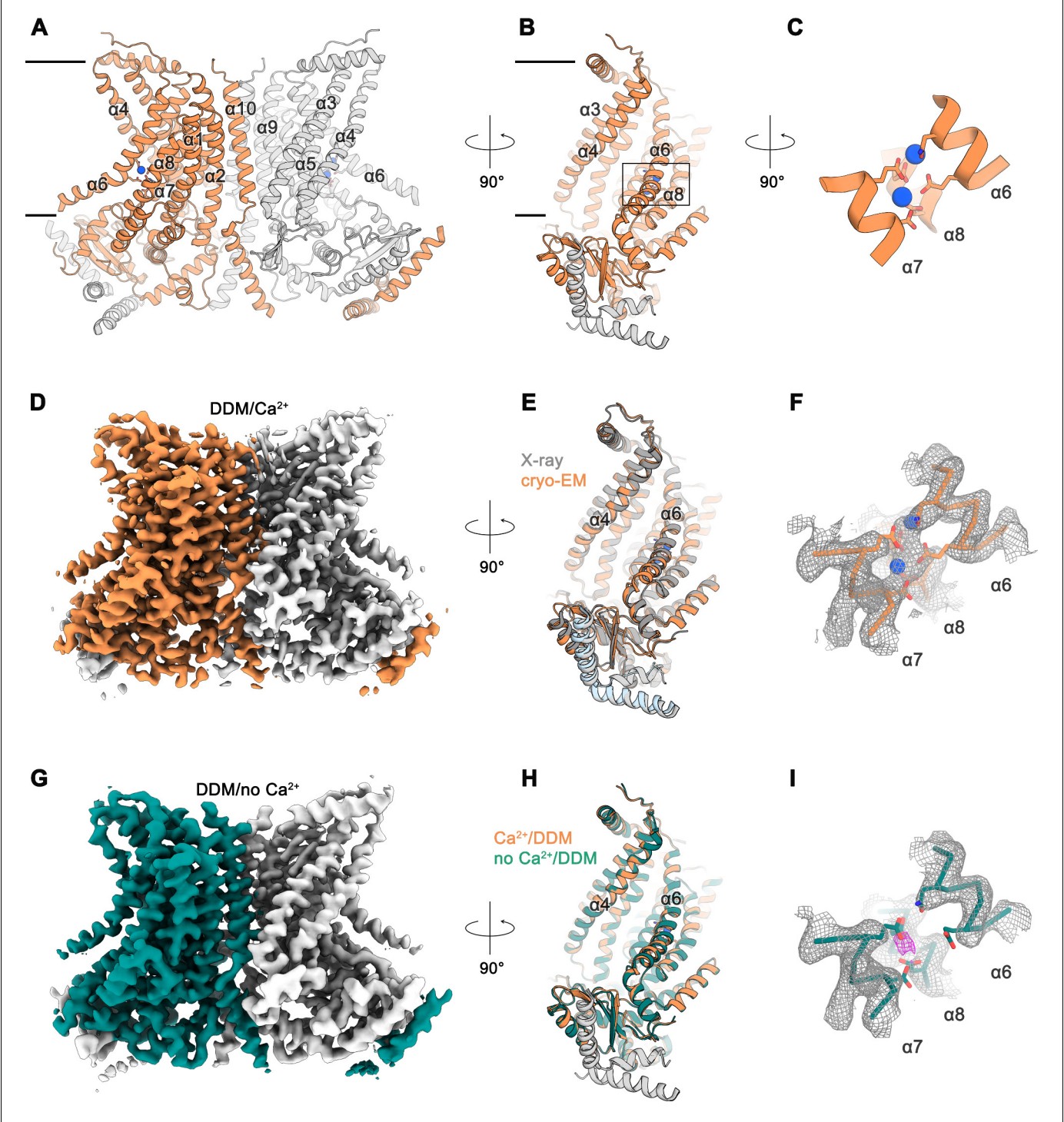

**Figure 1.** Cryo-EM structures of nhTMEM16 in detergent. Ribbon representation of the $Ca^{2+}$-bound cryo-EM structure of nhTMEM16 in detergent. The view is from within the membrane perpendicular to the long dimension of the protein (**A**), towards the subunit cavity (**B**) and at the $Ca^{2+}$-ion binding site (**C**). The membrane boundary is indicated. Subunits in the dimer are depicted in orange and gray, respective transmembrane-helices are labelled and $Ca^{2+}$-ions are displayed as blue spheres. Relative orientations are indicated and the location of the $Ca^{2+}$-binding site is highlighted by a box in panel B. (**D**) Cryo-EM map of the nhTMEM16 dimer (orange and gray) in DDM in presence of $Ca^{2+}$ at 3.6 Å, sharpened with a b-factor of –126 Å$^2$ and contoured at 6 σ. (**E**) Ribbon representation of a superposition of the $Ca^{2+}$-bound structure in detergent determined by cryo-EM (orange and gray) and the $Ca^{2+}$-bound X-ray structure (dark gray and light blue, PDBID: 4WIS) The view is as in panel B. (**F**) View of the $Ca^{2+}$-ion binding site of the $Ca^{2+}$-bound state of nhTMEM16 in DDM. $Ca^{2+}$-ions are displayed as blue spheres. (**G**) Cryo-EM map of the nhTMEM16 dimer (green and gray) in DDM in the

*Figure 1 continued on next page*

*Figure 1 continued*

absence of Ca$^{2+}$ at 3.7 Å, sharpened with a b-factor of –147 Å$^2$ and contoured at 6 σ. (H) Ribbon representation of a superposition of the Ca$^{2+}$-bound (orange and gray) and the Ca$^{2+}$-free structure (green and gray) in detergent determined by cryo-EM. The view is as in panel B. (I) View of the Ca$^{2+}$-binding site in the Ca$^{2+}$-free state of nhTMEM16 in DDM. The location of weak residual density at the center of the Ca$^{2+}$-binding site is indicated in magenta. F,I The respective cryo-EM densities are contoured at 7 σ and shown as mesh. The backbone is displayed as Cα-trace, selected side-chains as sticks.

DOI: https://doi.org/10.7554/eLife.44364.002

The following figure supplements are available for figure 1:

**Figure supplement 1.** Reconstitution of nhTMEM16 into liposomes.
DOI: https://doi.org/10.7554/eLife.44364.003

**Figure supplement 2.** Structure Determination of nhTMEM16 in DDM in complex with Ca$^{2+}$.
DOI: https://doi.org/10.7554/eLife.44364.004

**Figure supplement 3.** Structure Determination of nhTMEM16 in DDM in absence of Ca$^{2+}$.
DOI: https://doi.org/10.7554/eLife.44364.005

**Figure supplement 4.** Sequence alignment of selected TMEM16 scramblases.
DOI: https://doi.org/10.7554/eLife.44364.006

**Figure supplement 5.** Detergent binding to the subunit cavity.
DOI: https://doi.org/10.7554/eLife.44364.007

(*Figure 1F,I*). Due to the presence of millimolar amounts of EGTA in the buffer, the resulting free Ca$^{2+}$ concentration is in the pM range, well below the estimated nM potency of the divalent cation (*Figure 1—figure supplement 1*). Thus, whereas we do not want to exclude the possibility of Ca$^{2+}$ binding at low occupancy, this density might as well correspond to a bound Na$^+$ ion, which is present in the sample buffer at a concentration of 150 mM (*i.e.* at a 10$^{10}$ times higher concentration compared to Ca$^{2+}$). We think that the stability of the open conformation displayed in the structures of the protein in detergent, irrespectively of the presence of bound Ca$^{2+}$, does not reflect an intrinsic rigidity of the subunit cavity. Instead, it is likely a consequence of experimental conditions, which stabilize the observed conformation of the protein. This is in accordance with residual density from the headgroup of a bound detergent molecule at the upper part of the lipid translocation path found in both datasets, which might prevent subunit cavity closure (*Figure 1—figure supplement 5*). Interestingly, the detergent molecule is interacting with Tyr 439, a residue that was shown to be essential for lipid translocation in MD simulations and functional studies (*Lee et al., 2018*). The Ca$^{2+}$-free structure in detergent thus likely displays a conformation that is responsible for the basal activity observed for nhTMEM16 and other fungal scramblases (*Figure 1—figure supplement 1*) (*Brunner et al., 2014*; *Malvezzi et al., 2013*). We expect this state to be less populated in a membrane environment.

## Structure of the Ca$^{2+}$-free state of nhTMEM16 in lipid nanodiscs

The absence of conformational changes in the subunit cavity of nhTMEM16 in detergents motivated us to study the structure of the protein in a lipid bilayer. For that purpose, we have reconstituted nhTMEM16 into nanodiscs of different sizes using a lipid composition at which the protein retains its lipid scrambling activity, although with considerably slower kinetics (*Figure 1—figure supplement 1C*). In order to make sure that the scramblase is surrounded by a sufficiently large membrane area, we assembled 2N2 scaffolding proteins with nhTMEM16 at three different lipid to protein ratios (LPR) in the absence of Ca$^{2+}$ and collected small datasets for all of them (*Figure 2—figure supplement 1*). In all cases, the protein is located close to the center of an elliptic disc with its long dimension oriented about parallel to the shortest side of the disc, a behavior that is more pronounced in larger discs obtained at high lipid to protein ratios (*Figure 2—figure supplement 1*). The orientation reflects a preference of lipids for locations distant from the subunit cavity, which appears to deform the nanodisc. Since increasing the amount of lipids only caused elongation of the nanodisc in one direction without a change in its diameter near the subunit cavity, we proceeded with data collection on the sample with the lowest LPR of 145:1, which resulted in a homogeneous assembly of approximately 165 × 140 Å in size (*Figure 2—figure supplement 1A*). This approach was used to determine the structures of ligand-bound and ligand-free states of nhTMEM16 in a membrane-like environment. The data are of excellent quality and provide an unambiguous view of nhTMEM16 in

**Table 1.** Cryo-EM data collection, refinement and validation statistics.

| | nhTMEM16, DDM, +Ca$^{2+}$ (EMDB-4588, PDB 6QM5) | nhTMEM16, DDM, -Ca$^{2+}$ (EMDB-4589, PDB 6QM6) | nhTMEM16, 2N2, -Ca$^{2+}$ (EMDB-4587, PDB 6QM4) |
|---|---|---|---|
| **Data collection and processing** | | | |
| Microscope | FEI Talos Arctica | FEI Talos Arctica | FEI Talos Arctica |
| Camera | Gatan K2 Summit + GIF | Gatan K2 Summit + GIF | Gatan K2 Summit + GIF |
| Magnification | 49,407 | 49,407 | 49,407 |
| Voltage (kV) | 200 | 200 | 200 |
| Exposure time frame/total (s) | 0.15/9 | 0.15/9 | 0.15/9 |
| Number of frames per image | 60 | 60 | 60 |
| Electron exposure (e–/Å$^2$) | 52 | 52 | 52 |
| Defocus range (μm) | −0.5 to −2.0 | −0.5 to −2.0 | −0.5 to −2.0 |
| Pixel size (Å) | 1.012 | 1.012 | 1.012 |
| Box size (pixels) | 220 | 240 | 240 |
| Symmetry imposed | C2 | C2 | C2 |
| Initial particle images (no.) | 251,693 | 570,203 | 1,379,187 |
| Final particle images (no.) | 120,086 | 238,070 | 133,961 |
| Map resolution (Å) 0.143 FSC threshold | 3.64 | 3.68 | 3.79 |
| Map resolution range (Å) | 3.4–5.0 | 3.4–5.0 | 3.3–5.0 |
| **Refinement** | | | |
| Initial model used | PDB 4WIS | 6QM5 | 6QMB |
| Model resolution (Å) FSC threshold | 3.6 | 3.7 | 3.8 |
| Model resolution range (Å) | 15–3.6 | 15–3.7 | 15–3.8 |
| Map sharpening $B$ factor (Å$^2$) | −126 | −147 | −150 |
| **Model composition** | | | |
| Nonhydrogen atoms | 10874 | 10852 | 10578 |
| Protein residues | 1346 | 1344 | 1308 |
| Ligands | 4 | - | - |
| **$B$ factors (Å$^2$)** | | | |
| Protein | 55.75 | 76.42 | 94.09 |
| Ligand | 33.72 | - | - |
| **R.m.s. deviations** | | | |
| Bond lengths (Å) | 0.007 | 0.004 | 0.007 |
| Bond angles (°) | 0.821 | 0.835 | 0.983 |
| **Validation** | | | |
| MolProbity score | 1.47 | 1.42 | 1.53 |
| Clashscore | 4.02 | 4.35 | 4.09 |
| Poor rotamers (%) | 0 | 0 | 0.18 |
| **Ramachandran plot** | | | |
| Favored (%) | 95.92 | 96.74 | 95.17 |
| Allowed (%) | 4.08 | 3.26 | 4.83 |
| Disallowed (%) | 0 | 0 | 0 |

DOI: https://doi.org/10.7554/eLife.44364.008

different conformations (*Figures 2* and *3*; *Figure 2—figure supplement 2*; *Figure 3—figure supplements 1–2*; *Tables 1* and *2*).

The differences between structures of nhTMEM16 determined in detergents and in a lipid environment are most pronounced for the Ca$^{2+}$-free state determined at 3.8 Å resolution (*Figure 2A,B, D*; *Figure 2—figure supplement 2*). Whereas the structure in detergent displays a Ca$^{2+}$-free conformation where the subunit cavity remains open to the membrane, the same region has changed its accessibility in the nanodisc sample. Here, the movement of α4 leads to a shielding of the cavity

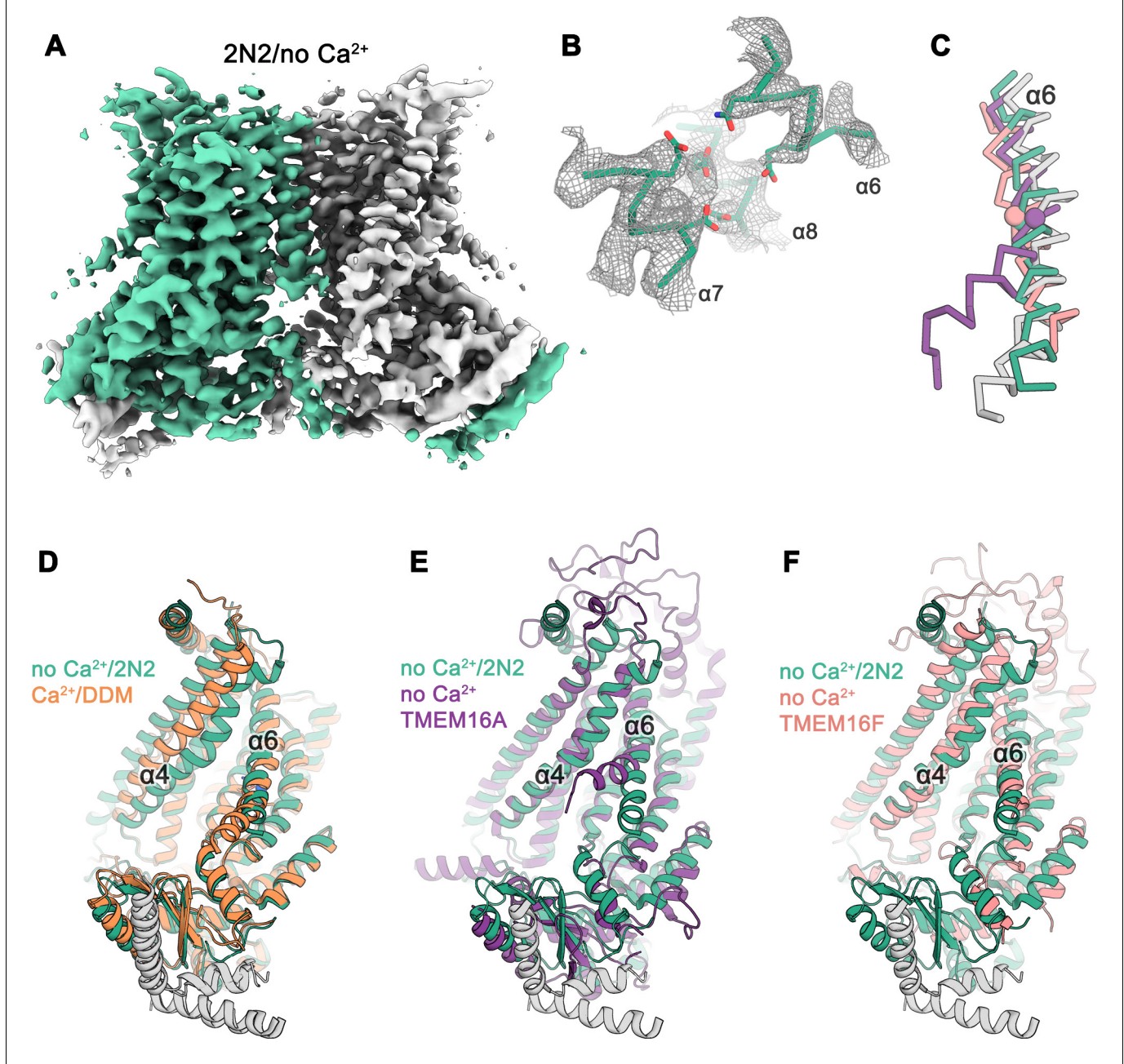

**Figure 2.** Cryo-EM structure of nhTMEM16 in nanodiscs in absence of $Ca^{2+}$. (**A**) Cryo-EM map of the nhTMEM16 dimer (light green and gray) in nanodiscs in absence of $Ca^{2+}$ at 3.8 Å resolution. The map was sharpened with a b-factor of −150 Å$^2$ and is contoured at 5.6 σ. (**B**) View of the $Ca^{2+}$-binding site in the $Ca^{2+}$-free state of nhTMEM16 in nanodiscs. The cryo-EM density shown as mesh is contoured at 6 σ, the backbone is displayed as Cα-trace and selected side-chains as sticks. (**C**) Cα-traces of α6 from a superposition of $Ca^{2+}$-free structures of nhTMEM16 in nanodiscs (green), TMEM16A (violet, PDBID: 5OYG) and TMEM16F (red, PDBID: 6QPB) and of the $Ca^{2+}$-bound structure of nhTMEM16 in DDM (gray). The spheres indicate the position of the flexible glycine residue in TMEM16A and TMEM16F, which acts as a hinge for conformational changes. (**D–F**) Ribbon representation of superpositions of the $Ca^{2+}$-free structure in nanodiscs (light green and gray) with the $Ca^{2+}$-bound structure in DDM ((**D**), orange and gray), the $Ca^{2+}$-free structure of TMEM16A ((**E**), violet, PDBID: 5OYG) and the $Ca^{2+}$-free structure of TMEM16F ((**F**), red, PDBID: 6QPB). Selected α-helices are labeled, the views are as in *Figure 1*.

DOI: https://doi.org/10.7554/eLife.44364.009

The following figure supplements are available for figure 2:

**Figure supplement 1.** Reconstitution of nhTMEM16 into nanodiscs.

DOI: https://doi.org/10.7554/eLife.44364.010

*Figure 2 continued on next page*

*Figure 2 continued*

**Figure supplement 2.** Structure Determination of nhTMEM16 in the absence of Ca$^{2+}$ in nanodiscs.
DOI: https://doi.org/10.7554/eLife.44364.011

from the membrane core, resembling a conformation observed in the ion channel TMEM16A (*Figure 2E*) (*Dang et al., 2017*; *Paulino et al., 2017a*; *Paulino et al., 2017b*). This movement is accompanied by a conformational change of α6 as a consequence of the repulsion of negatively charged residues in the empty Ca$^{2+}$-binding site (*Figure 2B–D*). Generally, the Ca$^{2+}$-free state of nhTMEM16 in nanodiscs is very similar to equivalent structures of its close homologs afTMEM16 (*Falzone et al., 2019*), TMEM16K (*Bushell et al., 2018*) and the more distantly related scramblase TMEM16F (*Alvadia et al., 2019*) (*Figure 2C,F* and *Figure 1—figure supplement 4*), thus underlining that all structures represent inactive conformations of the scramblase in the absence of ligand.

## Structures of Ca$^{2+}$-bound states of nhTMEM16 in lipid nanodiscs

Different from the data of the Ca$^{2+}$-free protein in nanodiscs, which comprises a single prevailing state, the cryo-EM data obtained for the Ca$^{2+}$-bound samples in the same environment shows a large heterogeneity along the 'subunit cavity'. This heterogeneity allows to sample the transition of the lipid permeation path from the closed to the open state and emphasizes the intrinsic dynamics of this region during activation (*Figure 3*; *Figure 3—figure supplements 1* and *2*; *Table 2*). A classification of the data yielded several states with distinct conformations of the 'subunit cavity', all of which exhibit well-resolved cryo-EM density for two Ca$^{2+}$ ions and a rigid arrangement of α6 in place to coordinate the divalent ions (*Figure 3A–C*; *Figure 3—figure supplements 1* and *2*). About 50% of the particles define classes where α3 and α4 are poorly resolved, thus emphasizing the high degree of flexibility in the region, which suggests that several states along the activation process might be transient (classes 5 and 6, *Figure 3—figure supplement 1C*). By contrast, three classes show well resolved distinct conformations. The 'open' class, at 3.6 Å resolution, encompasses about 25% of the particles and strongly resembles the Ca$^{2+}$-bound open state obtained in detergent, with a subunit cavity that is exposed to the membrane (*Figure 3A,D*; *Figure 3—figure supplements 1* and *2*). In contrast, the 'Ca$^{2+}$-bound closed' class determined from 15% of particles at 3.6 Å resolution, strongly resembles the 'Ca$^{2+}$-free closed' state obtained in nanodiscs, with a subunit cavity that is shielded from the membrane, except that Ca$^{2+}$ is bound and α6 is less mobile (*Figure 3C,G*; *Figure 3—figure supplements 1* and *2*). The third distinct state at 3.7 Å resolution encompasses 12% of all particles and shows an 'intermediate' conformation, where the subunit cavity is still shielded from the membrane but where a potential aqueous pore surrounded by α-helices of the subunit cavity has widened compared to the closed state (*Figure 3B,E,F*; *Figure 3—figure supplements 1* and *2*). The distribution of states, with a portion of nhTMEM16 residing in potentially non-conductive conformations correlates with the lower activity of the protein in this lipid composition (*Figure 1—figure supplement 1C*) and it emphasizes the equilibrium between conformations of nhTMEM6 in the Ca$^{2+}$-bound state. Unassigned weak density in the subunit cavity in the 'open' - but not in the 'closed' - conformation, hints at lipid molecules during the transition between the leaflets (*Figure 3—figure supplement 3*). Strikingly, the strongest density is located at an equivalent position as found for the potential DDM density in the cryo-EM maps of nhTMEM6 in detergent (*Figure 1—figure supplement 5*).

## Conformational transitions

The diversity of conformations of nhTMEM16 observed in various datasets and the presence of different populations in the Ca$^{2+}$-bound sample in lipid nanodiscs suggest that our structures represent distinct states along a stepwise activation process. Apart from a small rearrangements of the cytosolic domain, the core of the protein is virtually identical in all structures and the largest movements are observed in the vicinity of the subunit cavity (*Figure 2D*, *Figure 3D-G*, *Figure 4* and *Figure 5A* and *Video 1*). The Ca$^{2+}$-bound conformation in detergent and the 'open' class observed in the data of Ca$^{2+}$-bound protein in nanodiscs both show the widest opening of the subunit cavity and thus likely depict a scrambling-competent state of the protein (*Figure 4A*). In this open state, α-helices 4 and 6 are separated from each other on their entire length framing the two opposite edges

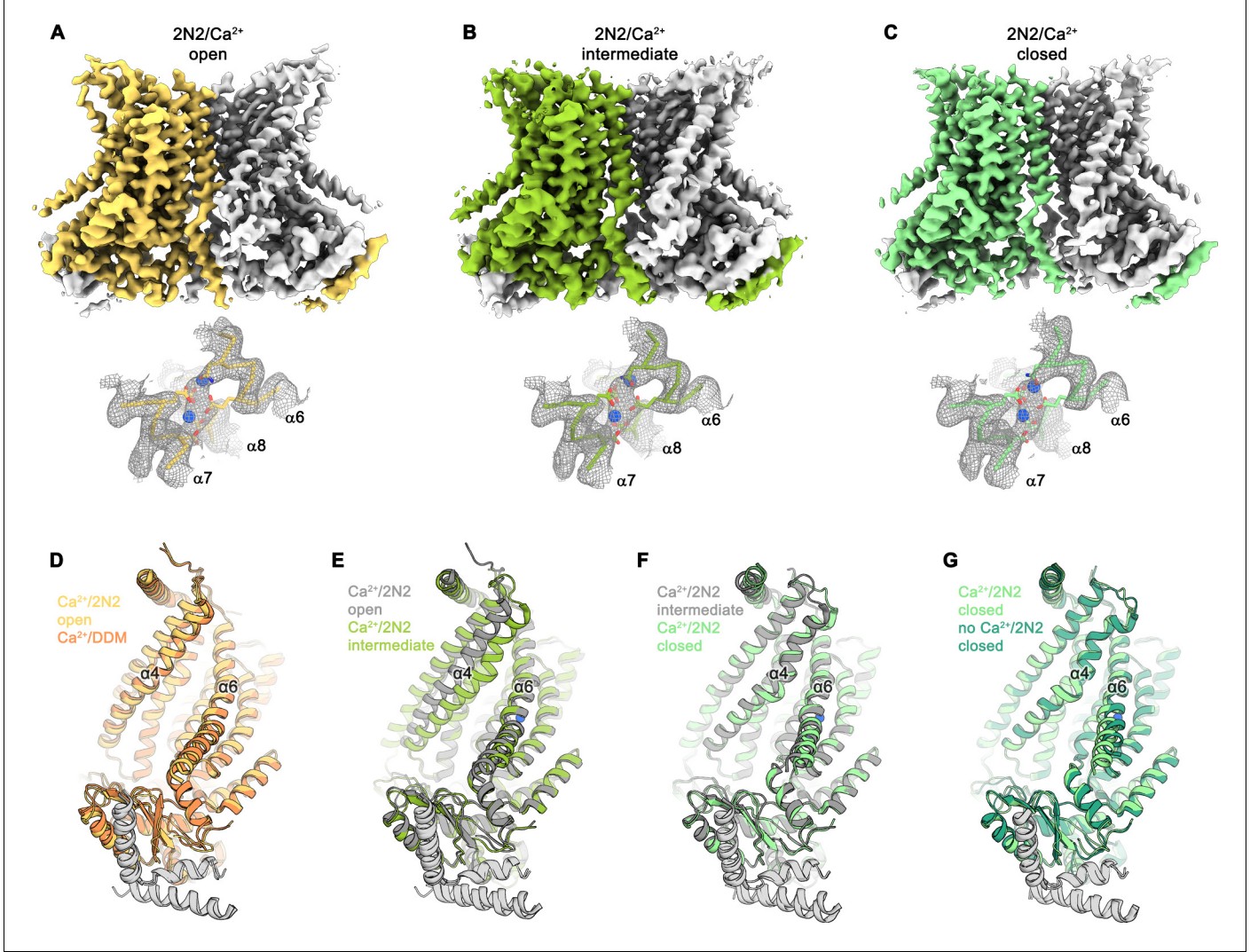

**Figure 3.** Cryo-EM structures of nhTMEM16 in nanodiscs in presence of $Ca^{2+}$. (**A**) Cryo-EM map of the nhTMEM16 dimer in the $Ca^{2+}$-bound 'open' state (yellow and gray) in nanodiscs at 3.6 Å, sharpened with a b-factor of –114 Å² and contoured at 6.5 σ (upper panel). Close-up of the $Ca^{2+}$-binding site with two bound $Ca^{2+}$-ions displayed as blue spheres (lower panel). Cryo-EM density contoured at 7 σ is shown as gray mesh, the backbone is displayed as yellow Cα-trace and selected side-chains as sticks. (**B**) Cryo-EM map of the nhTMEM16 dimer in the $Ca^{2+}$-bound 'intermediate' state (yellow-green and gray) in nanodiscs at 3.7 Å, sharpened with a b-factor of –96 Å² and contoured at 4.9 σ (upper panel). Close-up of the $Ca^{2+}$-binding site with two bound $Ca^{2+}$-ions displayed as blue spheres (lower panel). Cryo-EM density contoured at 7 σ is shown as gray mesh, the backbone is displayed as light green Cα-trace and selected side-chains as sticks. (**C**) Cryo-EM map of the nhTMEM16 dimer in the '$Ca^{2+}$-bound closed state (green and gray) in nanodiscs at 3.6 Å, sharpened with a b-factor of –96 Å² and contoured at 6.5 σ (upper panel). Close-up of the $Ca^{2+}$-binding site with two bound $Ca^{2+}$-ions displayed as blue spheres (lower panel). Cryo-EM density contoured at 7 σ is shown as gray mesh, the backbone is displayed as light green Cα-trace and selected side-chains as sticks. (**D–G**) Ribbon representation of superpositions with $Ca^{2+}$-bound structures of nhTMEM16 in nanodiscs: (**D**) $Ca^{2+}$-bound open structure in nanodiscs and $Ca^{2+}$-bound structure in DDM; (**E**) $Ca^{2+}$-bound open structure in nanodiscs and $Ca^{2+}$-bound intermediate structure in nanodiscs; (**F**) $Ca^{2+}$-bound intermediate structure in nanodiscs and $Ca^{2+}$-bound closed structure in nanodiscs; (**G**) $Ca^{2+}$-bound closed structure in nanodiscs and $Ca^{2+}$-free structure in nanodiscs. Selected helices are labeled and views are as in *Figure 1*.

DOI: https://doi.org/10.7554/eLife.44364.012

The following figure supplements are available for figure 3:

**Figure supplement 1.** Structure Determination of nhTMEM16 in complex with $Ca^{2+}$ in nanodiscs.

DOI: https://doi.org/10.7554/eLife.44364.013

**Figure supplement 2.** Conformational heterogeneity in the $Ca^{2+}$-bound data of nhTMEM16 in nanodiscs.

DOI: https://doi.org/10.7554/eLife.44364.014

**Figure supplement 3.** Lipid distribution in the subunit cavity.

DOI: https://doi.org/10.7554/eLife.44364.015

**Table 2.** Cryo-EM data collection, refinement and validation statistics of the $Ca^{2+}$ bound nhTMEM16 in nanodiscs.

| | nhTMEM16, 2N2, +$Ca^{2+}$open (EMDB-4592, PDB 6QM9) | nhTMEM16, 2N2, +$Ca^{2+}$intermediate closed (EMDB-4593, PDB 6QMA) | nhTMEM16, 2N2, +$Ca^{2+}$closed (EMDB-4594, PDB 6QMB) |
|---|---|---|---|
| Data collection and processing | | | |
| Microscope | FEI Talos Arctica | | |
| Camera | Gatan K2 Summit + GIF | | |
| Magnification | 49,407 | | |
| Voltage (kV) | 200 | | |
| Exposure time frame/total (s) | 0.15/9 | | |
| Number of frames per image | 60 | | |
| Electron exposure (e–/$Å^2$) | 52 | | |
| Defocus range (μm) | −0.5 to −2.0 | | |
| Pixel size (Å) | 1.012 | | |
| Box size (pixels) | 240 | | |
| Symmetry imposed | C2 | | |
| Initial particle images (no.) | 2,440,110 | | |
| Final particle images (no.) | 71,175 | 33,310 | 41,631 |
| Map resolution (Å) 0.143 FSC threshold | 3.57 | 3.68 | 3.57 |
| Map resolution range (Å) | 3.4–5 | 3.6–5 | 3.4–5 |
| Refinement | | | |
| Initial model used | 6QM5 | 6QMB | 6QM5 |
| Model resolution (Å) FSC threshold | 3.6 | 3.7 | 3.6 |
| Model resolution range (Å) | 15–3.6 | 15–3.7 | 15–3.6 |
| Map sharpening $B$ factor ($Å^2$) | −114 | −96 | −95 |
| Model composition Nonhydrogen atoms | 10878 | 10714 | 10696 |
| Protein residues | 1346 | 1324 | 1322 |
| Ligands | 4 | 4 | 4 |
| $B$ factors ($Å^2$) Protein | 77.93 | 84.86 | 88.53 |
| Ligand | 54.34 | 57.57 | 67.10 |
| R.m.s. deviations Bond lengths (Å) | 0.005 | 0.006 | 0.008 |
| Bond angles (°) | 0.880 | 0.872 | 0.858 |
| Validation MolProbity score | 1.36 | 1.38 | 1.42 |
| Clashscore | 3.74 | 4.69 | 3.90 |
| Poor rotamers (%) | 0 | 0.18 | 0 |
| Ramachandran plot Favored (%) | 96.82 | 97.23 | 96.38 |
| Allowed (%) | 3.18 | 2.77 | 3.62 |
| Disallowed (%) | 0 | 0 | 0 |

DOI: https://doi.org/10.7554/eLife.44364.016

of a semicircular polar furrow that is exposed to the lipid bilayer (*Figure 4A*). In this case, the interaction of α3 with α10 of the adjacent subunit on the intracellular side appears to limit the movement of α4, thus preventing a further widening of the cavity on the intracellular side (*Figure 5A-C*). On the opposite border of the subunit cavity, α6 is immobilized in its position by the bound $Ca^{2+}$ ions, which results in a tight interaction with α7 and α8 (*Figure 5D*). In the 'intermediate' class found in the $Ca^{2+}$-bound sample in nanodiscs, we find marked conformational changes compared to the 'open state'. The most pronounced differences are found for the α3-α4 pair. The intracellular part of α3 (but not the α2-α3 loop contacting the helix Cα1 following α10) has detached from its interaction

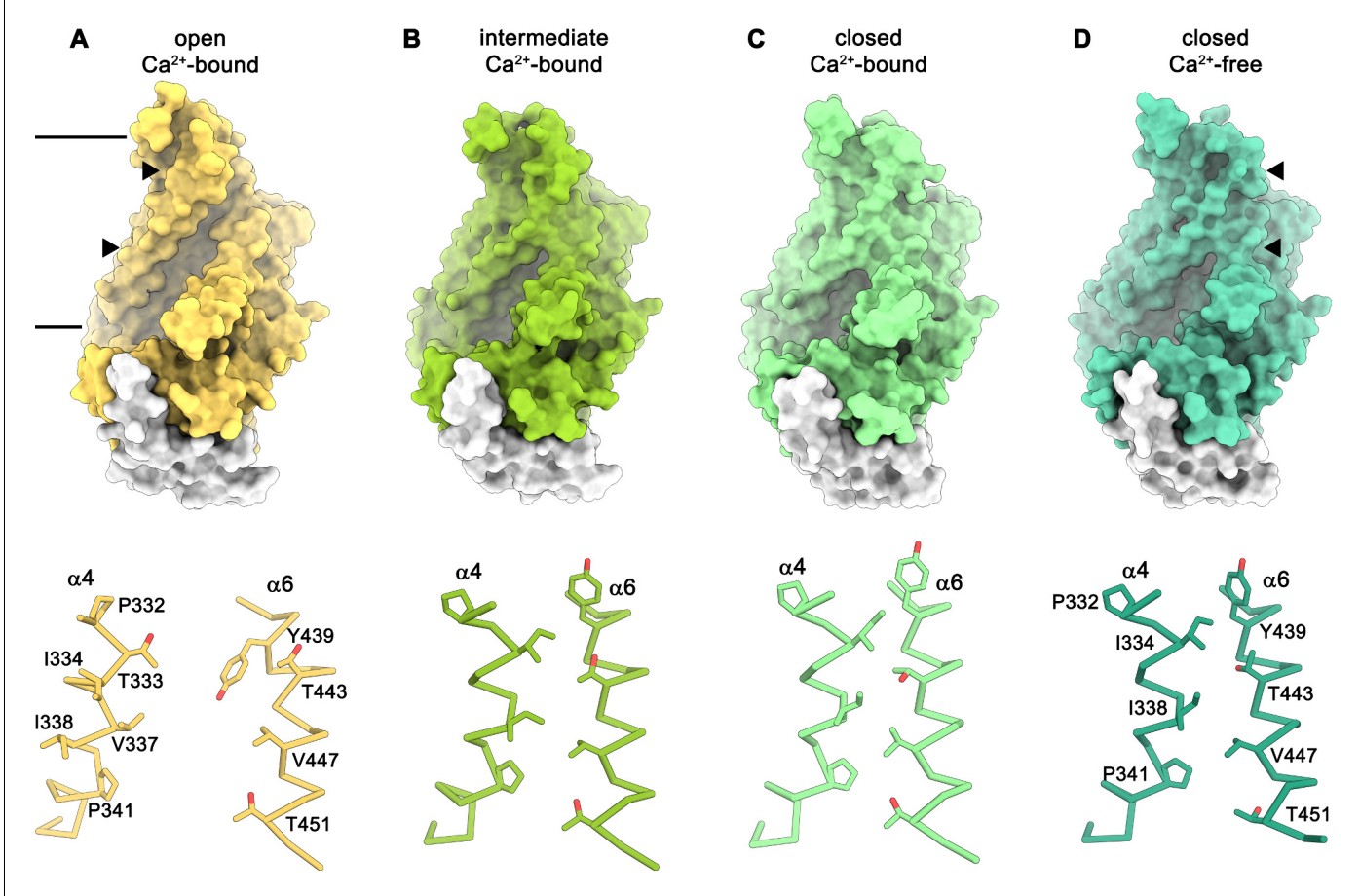

**Figure 4.** Conformations of the subunit cavity. Views of the molecular surface of the subunit cavity (top) and structures of α4 and α6 that line the cavity, displayed as Cα-trace with selected sidechains shown as sticks and labeled (bottom). (**A**) 'Open state' as defined in the Ca²⁺-bound structure in nanodiscs. (**B**) 'Intermediate state' as defined in the Ca²⁺-bound structure in nanodiscs, (**C**) 'Ca²⁺-bound closed state' as defined in the Ca²⁺-bound structure in nanodiscs. (**D**) 'Closed state' as defined by the Ca²⁺-free structure in nanodiscs. Membrane boundaries are indicated and the region shown in the lower panels is depicted by triangles in the upper panels.
DOI: https://doi.org/10.7554/eLife.44364.017

with α10, leading to a concomitant movement of α4 at the same region (*Figure 5B*). This movement has also affected the conformation of the extracellular part of α4, which has rotated and approached α6, thereby forming initial contacts between both helices (*Figures 4B and Figure 5C,D*). The subunit cavity in this conformation is no longer exposed to the membrane at its extracellular part and the structure likely represents an intermediate of the scramblase towards activation (*Figure 4B*). This 'intermediate state' is remarkable in light of the observed ion conduction of nhTMEM16 and other TMEM16 scramblases (*Alvadia et al., 2019*; *Lee et al., 2016*; *Suzuki et al., 2013*; *Yang et al., 2012*; *Yu et al., 2015*), as it might harbor a potential protein-enclosed aqueous pore located in a region that is expanded compared to the 'closed state' (*Figure 5—figure supplement 1*). The interactions between α4 and α6 on the extracellular side further tighten in the 'closed' class found in the Ca²⁺-bound sample in nanodiscs, thereby constricting the aqueous pore observed in the 'intermediate state' (*Figure 4C*; *Figure 5—figure supplement 1*). This conformation strongly resembles the Ca²⁺-free 'closed' conformation observed in nanodiscs, except that in the latter, the dissociation of the ligand from the protein has allowed a relaxation of α6 leading to its detachment from α8 at the intracellular part of the helix (*Figures 4D and Figure 5C,D*). Although, unlike TMEM16A and TMEM16F, nhTMEM16 does not contain a flexible glycine residue at the hinge, the movements of α6 upon Ca²⁺-release occur at a similar region (*Alvadia et al., 2019*; *Paulino et al., 2017a*) (*Figure 2C–F* and *Figure 1—figure supplement 4*). In the transition from an 'open' to a 'closed

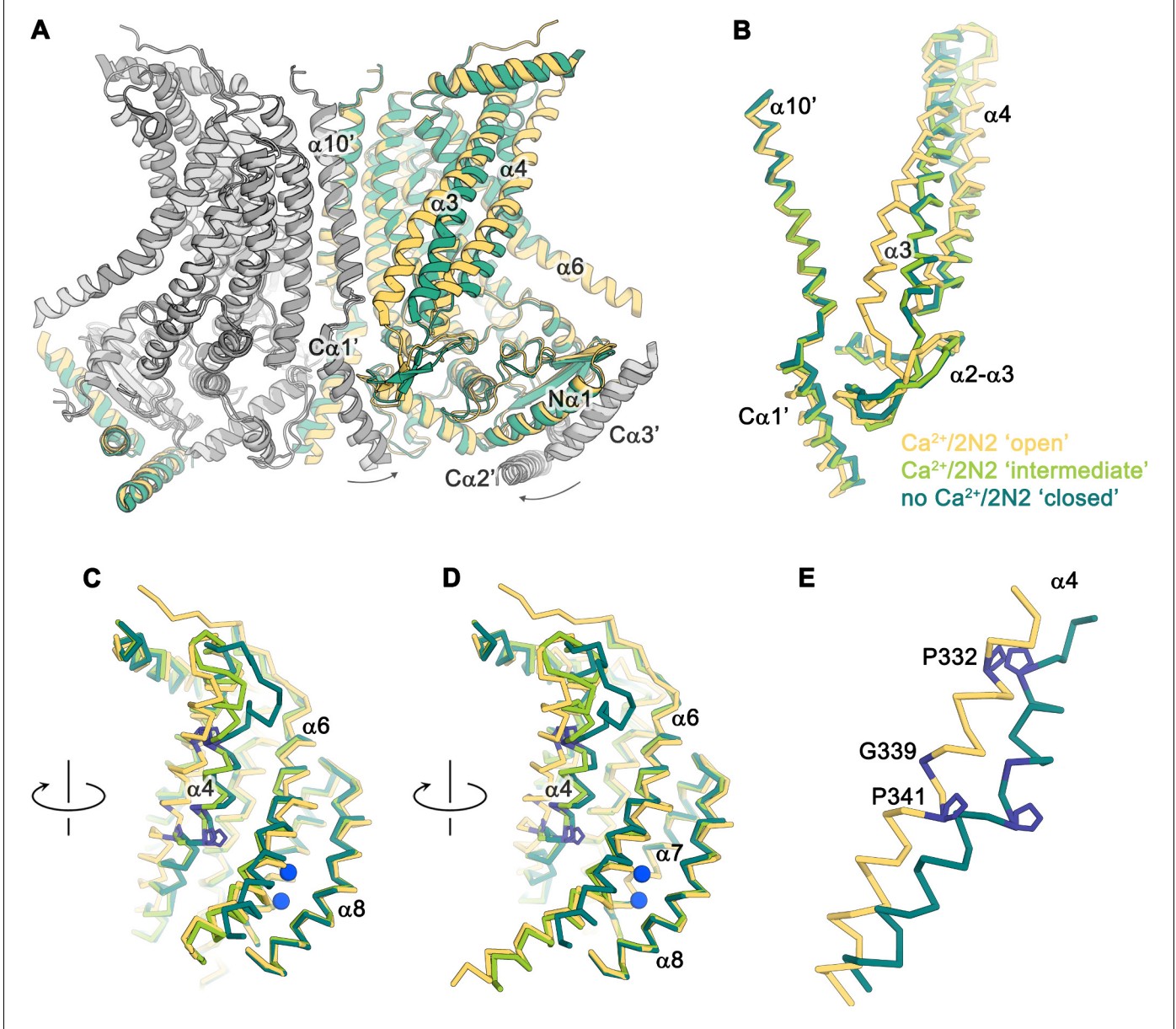

**Figure 5.** Conformational changes. (A) Ribbon representation of a superposition of the $Ca^{2+}$-bound 'open' (yellow and light gray) and $Ca^{2+}$-free 'closed' (green and dark gray) nhTMEM16 structures in nanodiscs. Selective helices are labeled and the arrows indicate small rearrangements of the cytosolic domain upon $Ca^{2+}$ release. (B–D) Cα-traces of selected regions of the 'subunit cavity' in different conformations as defined by the $Ca^{2+}$-bound 'open' structure in nanodiscs (yellow), the $Ca^{2+}$-bound 'intermediate' structure in nanodiscs (light green) and the $Ca^{2+}$-free 'closed' structure in nanodiscs (dark green). The view in B is as in A and the respective orientations of subsequent panels are indicated. (E) Superposition of α4 in the 'open' and 'closed' states with residues that act as potential pivots for structural arrangements highlighted in blue.

DOI: https://doi.org/10.7554/eLife.44364.018

The following figure supplement is available for figure 5:

**Figure supplement 1.** Diameter of the subunit cavity in $Ca^{2+}$-bound nhTMEM16 structures in nanodiscs.

DOI: https://doi.org/10.7554/eLife.44364.019

state', α4 undergoes the largest changes among all parts of nhTMEM16 with two prolines (Pro 332 and Pro 341) and a glycine (Gly 339) serving as potential pivots for helix-rearrangements (*Figure 5E* and *Video 1*). Remarkably, both prolines are conserved among fungal homologues and Pro 332 is also found in TMEM16K (*Bushell et al., 2018*), the closest mammalian orthologue of nhTMEM16 (*Figure 1—figure supplement 4*).

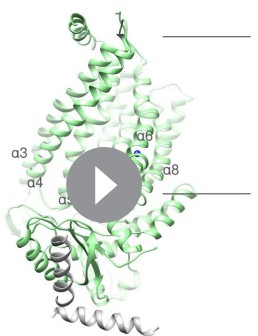

**Video 1.** Conformational transitions. Shown is a morph between distinct conformational states obtained for nhTMEM16, namely: the $Ca^{2+}$-bound 'open' state obtained in nanodiscs; the $Ca^{2+}$-bound 'intermediate' state obtained in lipid nanodiscs; the $Ca^{2+}$-bound 'closed' state obtained in lipid nanodiscs; and the $Ca^{2+}$-free closed state obtained in lipid nanodiscs. Shown is a view of the subunit cavity as depicted in *Figure 3* and *4*. $Ca^{2+}$ ions are shown as blue spheres. DOI: https://doi.org/10.7554/eLife.44364.020

## Protein-induced distortion of the lipid environment

A remarkable feature observed in the cryo-EM maps concerns the arrangement of detergent or lipid molecules surrounding nhTMEM16, which allowed us to characterize the influence of the protein on its environment. The observed distortions are found to a similar extent in all determined structures, irrespective of the presence or absence of calcium in detergent or lipid nanodiscs and can thus be considered a state-independent property of the protein. Neither detergent molecules nor nanodisc lipids are uniformly distributed on the outside of nhTMEM16 (*Figure 6* and *Video 2*). Instead, they adapt to the shape of the protein with distortions observed in the vicinity of the shorter α-helices 1 and 8 at the extracellular side and at the gap between α-helices 4 and 6 at the intracellular side (*Figure 6*). These distortions result in a marked deviation from the annular shape of detergents molecules or lipids surrounding the protein found in most membrane proteins. Remarkably, the resulting undulated distribution contains depressions in both the detergent and lipid density close to the entrances of the subunit

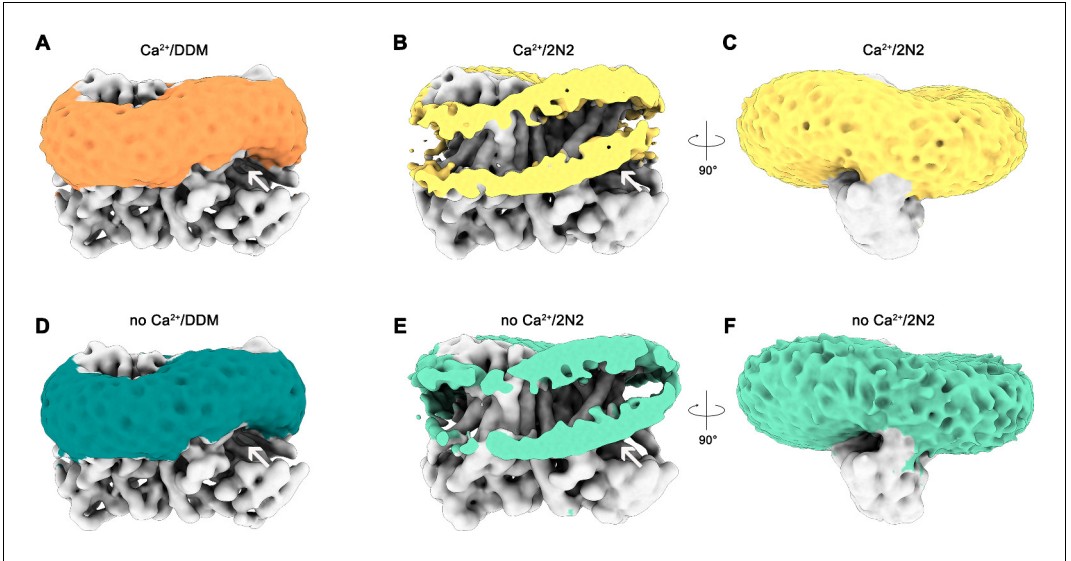

**Figure 6.** Detergent and lipid interactions. Shown are refined and unmasked cryo-EM density maps low-pass filtered to 6 Å. (**A**) Map of the $Ca^{2+}$-bound nhTMEM16 data in detergent contoured at 4 σ; (**B,C**) Map of the $Ca^{2+}$-bound nhTMEM16 data in nanodiscs contoured at 3.2 and 1.6 σ; (**D**) Map of the $Ca^{2+}$-free nhTMEM16 data in detergent contoured at 4 σ; (**E and F**), Map of the $Ca^{2+}$-free nhTMEM16 data in nanodiscs contoured at 3.2 and 1.6 σ. The density corresponding to the detergent micelle or the nanodisc, which is composed of lipids surrounded by the 2N2 belt protein, are colored in orange, yellow, dark green and light green, respectively, density of nhTMEM16 is shown in gray. A,B,D,E show a front view of the dimer similar to *Figure 1A* and the subunit cavity is indicated by a white arrow. B, E, clipped maps reveal the headgroup regions of both membrane leaflets. C,F, are viewed in direction of the subunit cavity.
DOI: https://doi.org/10.7554/eLife.44364.021

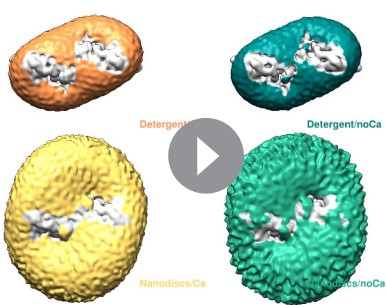

**Video 2.** Micelle and lipid distortion. Shown are the cryo-EM density maps of nhTMEM16 obtained in detergent in complex with $Ca^{2+}$ (upper left corner), in detergent in absence of $Ca^{2+}$ (upper right corner), in lipid nanodiscs in complex with $Ca^{2+}$ (lower left corner) and in lipid nanodiscs in absence of $Ca^{2+}$ (lower right corner). The surrounding environment corresponding to the detergent micelle (upper panels) or the lipid nanodiscs (lower panels) are colored respectively, while the rest of the protein is shown in gray. Refined and unmasked cryo-EM maps were low-pass filtered to 6 Å. Transmembrane α-helices 4 and 6 are labeled at the end, indicating the position of the subunit cavity.
DOI: https://doi.org/10.7554/eLife.44364.022

cavity (*Figure 6* and *Video 2*). The arrangement of the protein with its long dimension parallel to the short diameter of the oval-shaped nanodiscs reflects the preferential location of lipids at sites distant from the subunit cavity (*Figure 2—figure supplement 1*). The protein affects the distribution of lipids and the shape of the plane of the bilayer in nanodiscs, which is slightly V-shaped and locally deviates from planarity at several regions (*Figure 6B,C,E,F*). These pronounced depressions thin and distort the lipid bilayer structure at both ends of the subunit cavity, thereby likely facilitating the entry of polar headgroups into the cavity and lowering the barrier for lipid movement (*Figure 6B,C,E,F*).

## Functional properties of mutants affecting the activation process

The distinct conformations of nhTMEM16 determined in this study hint at a sequential mechanism, in which $Ca^{2+}$-dependent and independent steps are coupled to promote activation similar to ligand-dependent ion channels. We have thus investigated the functional consequences of mutations of residues of the $Ca^{2+}$-binding site and of the hinges for α4 movement with our liposome-based lipid scrambling assay. The mutation D503A located on α7 concerns a residue of the $Ca^{2+}$-binding site that does not change its position in different protein conformations and thus should affect the initial ligand-binding step (*Figure 7A*). In this case, we find a $Ca^{2+}$-dependent scrambling activity, although with strongly decreased potency of the ligand (*Figure 7B,C*). Next, we investigated whether the mutation of two proline residues and a close-by glycine (Pro 332, Gly 339 and Pro 341), which form potential pivots during the rearrangement of α4 would affect lipid scrambling (*Figure 7A*). Whereas we did not find any detectable effect in the mutant P341A, the mutants G339A and P332A showed a strongly decreased activity both in the presence and absence of $Ca^{2+}$, but no detectable change in the $Ca^{2+}$-potency of the protein *Figure 7D-F*; *Figure 7—figure supplement 1*). As the residues do not face the subunit cavity in the 'open state' and are unlikely to interact with translocating lipids, we assume that these mutants alter the equilibrium between active and inactive conformations thus emphasizing the role of conformational changes in α4 for nhTMEM16 activation.

## Discussion

By combining data from cryo-electron microscopy and biochemical assays, our study has addressed two important open questions concerning the function of a TMEM16 scramblase: it has revealed the conformational changes that lead to the activation of the protein in response to $Ca^{2+}$ binding and it has shown how the protein interacts with the surrounding membrane to facilitate lipid translocation. Structures in $Ca^{2+}$-free and bound states reveal distinct conformations of nhTMEM16, which capture the catalytic subunit cavity of the scramblase at different levels of exposure to the membrane during the activation process, representing a 'closed', at least one 'intermediate' and an 'open' state. As different protein conformations are observed in presence and absence of its ligand, we propose a stepwise activation mechanism for TMEM16 scramblases, where all states are at equilibrium (*Figure 8*; *Figure 8—figure supplement 1*). Here, the $Ca^{2+}$-free conformation obtained in nanodiscs defines a non-conductive state of the protein, where the polar subunit cavity is shielded from the membrane by tight interactions of α4 and α6 on the extracellular part of the membrane (*Figure 4D*). The protein surface of the closed subunit cavity is hydrophobic as expected for a membrane protein. This 'closed state' of nhTMEM16 resembles equivalent structures obtained for the closely related

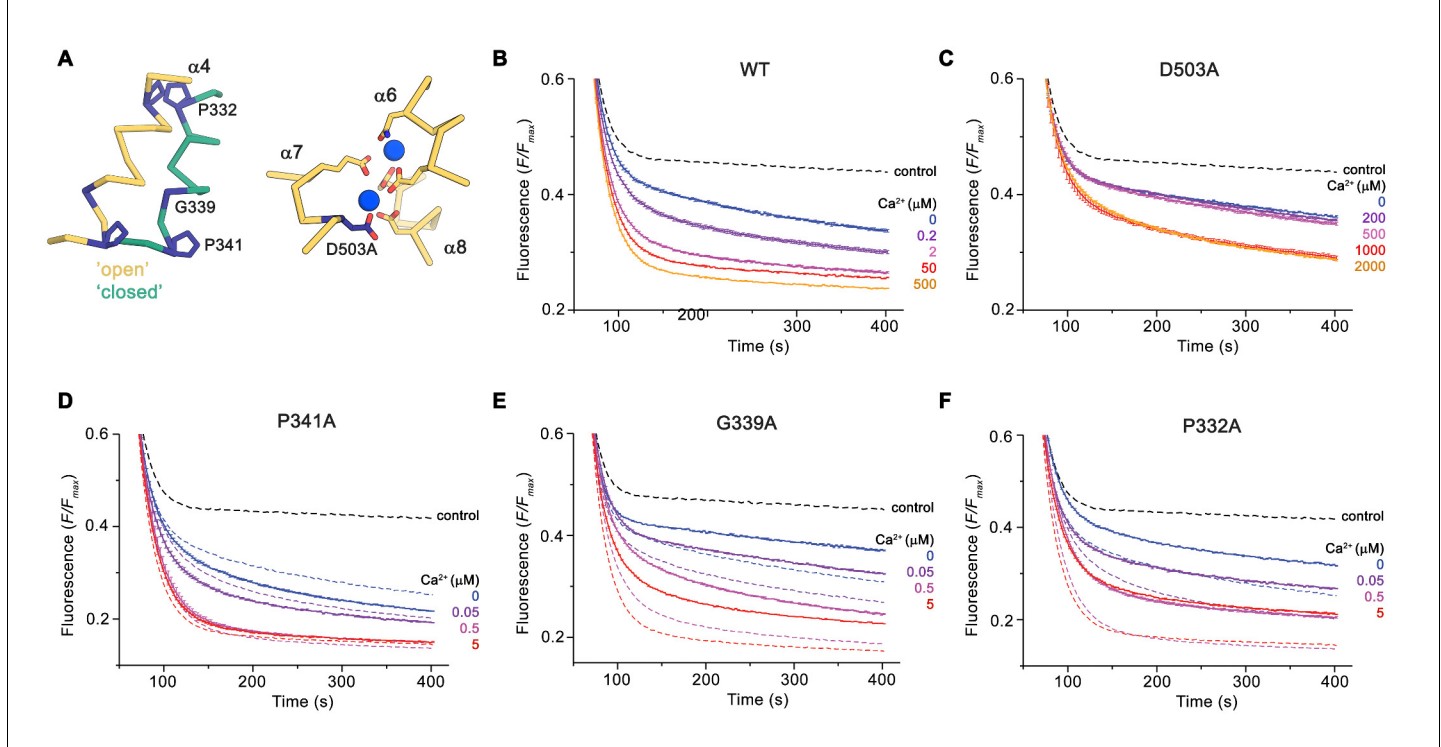

**Figure 7.** Functional properties of mutants. (A) Regions of nhTMEM16 harboring mutated residues. Left, Cα-trace of α4 in 'open' (yellow) and 'closed' (green) conformations of Ca²⁺-bound nhTMEM16 in nanodiscs. Right, ion binding site with Ca²⁺ ions shown in blue. (B–F) Ca²⁺-dependence of scrambling activity in nhTMEM16-containing proteoliposomes. Traces depict sections of the fluorescence decrease of tail-labeled NBD-PE lipids after addition of dithionite at different Ca²⁺ concentrations. Data show averages of three technical replicates, errors are s.e.m.. Ca²⁺ concentrations (µM) are indicated. Full traces are displayed in *Figure 7—figure supplement 1*. (B) WT, (C) D503A, (D) P341A, (E) G339A, and (F) P332A. (D to F) The traces of WT reconstituted in the same batch of liposomes are shown as dashed lines in the same color as their corresponding conditions of the mutant for comparison. (B to F), The black dashed line refers to the fluorescence decay in protein-free liposomes.
DOI: https://doi.org/10.7554/eLife.44364.023

The following figure supplement is available for figure 7:

**Figure supplement 1.** Reconstitution efficiency.
DOI: https://doi.org/10.7554/eLife.44364.024

lipid scramblases afTMEM16 (*Falzone et al., 2019*), TMEM16K (*Bushell et al., 2018*), the more distantly related TMEM16F (*Alvadia et al., 2019*) and the ion channel TMEM16A (*Figure 2E,F*) (*Dang et al., 2017*; *Paulino et al., 2017a*; *Paulino et al., 2017b*), thus underlining that it is representative for an inactive conformation of both functional branches of the family. In this conformation, the Ca²⁺-binding site is empty, with the intracellular part of α6 positioned apart from the remainder of the binding site to release the electrostatic strain induced by the large negative net-charge in this region. Details of the conformation of α6 in the absence of Ca²⁺ differ between the family members of known structure with the conformation in nhTMEM16 being close to the one observed for TMEM16F (*Alvadia et al., 2019*; *Paulino et al., 2017a*).

During the activation process, Ca²⁺ binding provides interactions with residues on α6. The helix rearranges into a locked state, as manifested in the well resolved cryo-EM density for its intracellular half, which shields the binding site from the cytoplasm. This Ca²⁺-induced movement of α6 in nhTMEM16 is coupled to rearrangements in its interface with α4, potentially weakening the interactions and resulting in a widening of the cavity. A cavity with a changed accessibility to the membrane is displayed in a population of particles observed in the Ca²⁺-bound nhTMEM16 structure in nanodiscs (*Figure 4B*). As the subunit cavity is not yet exposed to the membrane, the 'intermediate state' likely does not promote lipid scrambling. Instead, it might have opened a protein-enclosed pore that could facilitate ion conduction (*Figure 5—figure supplement 1*), which has been described as

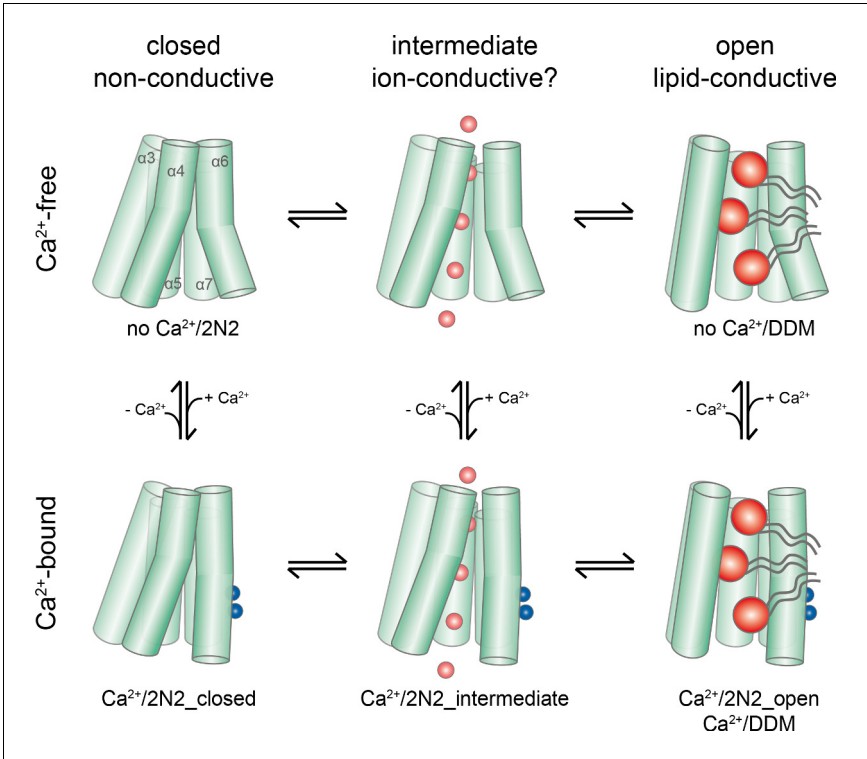

**Figure 8.** Activation mechanism. Scheme of the stepwise activation of nhTMEM16 displaying the equilibrium of states in $Ca^{2+}$-bound and $Ca^{2+}$-free conditions. Conformations obtained in this study and their correspondence to distinct states are indicated. $Ca^{2+}$ and permeating ions are depicted as blue and red spheres, respectively. Phospholipid headgroups are shown as red spheres and acyl chains as gray tails.

DOI: https://doi.org/10.7554/eLife.44364.025

The following figure supplement is available for figure 8:

**Figure supplement 1.** Activation mechanism.

DOI: https://doi.org/10.7554/eLife.44364.026

by-product of lipid scrambling in nhTMEM16 (*Lee et al., 2016*) and which is a hallmark of the lipid scramblase TMEM16F (*Alvadia et al., 2019*; *Yang et al., 2012*; *Yu et al., 2015*). Following the initial $Ca^{2+}$-induced conformational transition, the activation of lipid scrambling requires a second step, which leads to a larger reorientation of α-helices 3 and 4 and a subsequent opening of the polar subunit cavity to the membrane (*Figures 3* and *4*). This open state is defined by the $Ca^{2+}$-bound structure of nhTMEM16 obtained in detergent and by one of the classes observed in the $Ca^{2+}$-bound nanodisc dataset.

Thus, ion conduction and lipid-scrambling in nhTMEM16 and other TMEM16 scramblases might be mediated by distinct conformations which are at equilibrium in a $Ca^{2+}$-bound protein ( alternating pore/cavity mechanism, *Figure 8*; *Figure 8—figure supplement 1*). This stepwise activation mechanism is in general accordance with molecular dynamics simulations, which have started from the fully $Ca^{2+}$-bound open structure and for which transitions leading to a partial closure of the subunit cavity have been observed in the trajectories (*Jiang et al., 2017*; *Lee et al., 2018*). The basal scrambling activity of nhTMEM16 is likely a consequence of the equilibrium between open, intermediate and closed conformations, which is strongly shifted towards the closed state in the absence of $Ca^{2+}$. This is consistent with the observation of a $Ca^{2+}$-free open state in detergent but not in nanodiscs, where the detergent environment favors the equilibrium towards the open conformation (*Figure 1H*).

The cryo-EM structures of nhTMEM16 also reveal how the protein interacts with the surrounding membrane to disturb its structure and consequently lower the energy barrier for lipid flip-flop, as predicted from molecular dynamics simulations (*Bethel and Grabe, 2016*; *Jiang et al., 2017*; *Lee et al., 2018*; *Stansfeld et al., 2015*). Our data reveal a pronounced distortion of the protein-

surrounding environment, which is stronger than observed for the ion channel TMEM16A (*Paulino et al., 2017b*). This effect is preserved in all four datasets, irrespectively of the presence or absence of ligand in detergent or nanodiscs, as their underlying structural features change little in the different conformations. The distortion causes a deviation of the detergent or lipid molecules surrounding nhTMEM16 from the annular structure observed in most membrane proteins (*Figure 6* and *Video 2*). The resulting deformation of the membrane at the entrances to the subunit cavity potentially serves to channel lipid headgroups through this hydrophilic furrow on their way across the membrane.

In summary, our structures of nhTMEM16 have described how a calcium-activated lipid scramblase is activated in a stepwise manner and how it bends the surrounding bilayer to promote the transport of lipids between both leaflets of the membrane by a mechanism that is likely conserved within the family.

# Materials and methods

## Key resources table

| Reagent type (species) or resource | Designation | Source or reference | Identifiers | Additional information |
|---|---|---|---|---|
| Antibody | Mouse monoclonal Anti-c-Myc | Millipore Sigma | Cat#M4439; Clone#9E10 | (1:5000) |
| Antibody | Peroxidase Affinipure goat anti-mouse IgG | Jackson Immunoresearch | Cat#115-035-146 | (1:10000) |
| Chemical compound, drug | *n*-dodecyl-β-d-maltopyranoside (DDM), Solgrade | Anatrace | Cat#D310S | |
| Chemical compound, drug | Water for molecular biology | Millipore | H20MB1006 | |
| Chemical compound, drug | Calcium nitrate tetrahydrate | Millipore Sigma | Cat#C4955 | |
| Chemical compound, drug | Sodium chloride | Millipore Sigma | Cat#71380 | |
| Chemical compound, drug | Magnesium chloride | Fluka | Cat#63065 | |
| Chemical compound, drug | HEPES | Millipore Sigma | Cat#H3375 | |
| Chemical compound, drug | Ethylene glycol-bis (2-aminoethylether)-N,N,N′,N′-tetraacetic acid | Millipore Sigma | Cat#03777 | |
| Chemical compound, drug | Ethylenediamin etetraacetic acid | Millipore Sigma | Cat#E6758 | |
| Chemical compound, drug | cOmplete, EDTA-free Protease Inhibitor Cocktail | Roche | Cat#5056489001 | |
| Chemical compound, drug | Biotin | Millipore Sigma | Cat#B4501 | |
| Chemical compound, drug | D-desthiobiotin | Millipore Sigma | Cat#D1411 | |
| Chemical compound, drug | Glycerol | Millipore Sigma | Cat#G5516 | |
| Chemical compound, drug | 1-palmitoyl-2-oleoyl-glycero-3-phosphocholine | Avanti Polar Lipids, Inc | Cat#850457C | |
| Chemical compound, drug | 1-palmitoyl-2-oleoyl -sn-glycero-3-phospho-(1′-rac-glycerol) | Avanti Polar Lipids, Inc | Cat#840457C | |
| Chemical compound, drug | Diethyl ether | Millipore Sigma | Cat#296082 | |

*Continued on next page*

Continued

| Reagent type (species) or resource | Designation | Source or reference | Identifiers | Additional information |
|---|---|---|---|---|
| Chemical compound, drug | *E. coli* polar lipid extract | Avanti Polar Lipids, Inc | Cat#100600C | |
| Chemical compound, drug | Egg PC, 95% | Avanti Polar Lipids, Inc | Cat#131601C | |
| Chemical compound, drug | 18:1-06:0 NBD-PE | Avanti Polar Lipids, Inc | Cat#810155C | |
| Chemical compound, drug | Potassium chloride | Millipore Sigma | Cat#746436 | |
| Chemical compound, drug | CHAPS | Millipore Sigma | Cat#C3023 | |
| Chemical compound, drug | Sodium dithionite | Millipore Sigma | Cat#157953 | |
| Chemical compound, drug | Yeast nitrogen base without amino acids | Millipore Sigma | Cat#Y0626 | |
| Chemical compound, drug | Yeast Synthetic Drop-out Medium Supplements without uracil | Millipore Sigma | Cat#Y1501 | |
| Chemical compound, drug | D-Glucose | Applichem | Cat#A1422 | |
| Chemical compound, drug | Galactose | Millipore Sigma | Cat#G0625 | |
| Chemical compound, drug | Lithium acetate | Millipore Sigma | Cat#L4158 | |
| Chemical compound, drug | PEG3350 | Millipore Sigma | Cat#88276 | |
| Peptide, recombinant protein | DNAse I | AppliChem | A3778-0500 | |
| Peptide, recombinant protein | HRV 3C protease | Raimund Dutzler laboratory | | |
| Commercial assay or kit | Pierce Streptavidin Plus UltraLink Resin | Thermo Fisher Scientific | Cat#53117 | |
| Commercial assay or kit | Bio-Beads SM-2 Adsorbents | Bio-Rad | Cat# 1523920 | |
| Commercial assay or kit | Amicon Ultra-4–100 KDa cutoff | EMD Millipore | Cat#UFC8100 | |
| Commercial assay or kit | 0.22 µm Ultrafree-MC Centrifugal Filter | EMD Millipore | Cat#UFC30GV | |
| Commercial assay or kit | Strep-Tactin Superflow high capacity | IBA Lifesciences | Cat#2-1208-010 | |
| Commercial assay or kit | Superdex 200 10/300 GL | GE Healthcare | Cat# 17-5175-01 | |
| Commercial assay or kit | Superose 6 10/300 GL | GE Healthcare | Cat#17-5172-01 | |
| Strain, strain background (S. cerevisiae) | FGY217 | David Drew laboratory | | |
| Recombinant DNA reagent | nhTMEM16 open reading frame | GenScript | PubMed accession number XM_003045982 | |

*Continued on next page*

*Continued*

| Reagent type (species) or resource | Designation | Source or reference | Identifiers | Additional information |
|---|---|---|---|---|
| Recombinant DNA reagent | Yeast expression vector with N-terminal streptavidin binding peptide, Myc tag and 3C protease cleavage site | Raimund Dutzler laboratory | | |
| Recombinant DNA reagent | Membrane scaffold protein (MSP) 2N2 | Stephen Sligar laboratory | Addgene:Cat#29520 | |
| Software, algorithm | Online WEBMAXC calculator | *Bers et al., 2010* | http://maxch elator.stanford .edu/webmaxc/ webmaxcS.htm | |
| Software, algorithm | *Focus 1.1.0* | *Biyani et al. (2017)* | https://focus.c-cina. unibas.ch/about.php | |
| Software, algorithm | MotionCorr2 1.1.0 | *Zheng et al. (2017)* | http://msg. ucsf.edu/em /software/ motioncor2.html | |
| Software, algorithm | CTFFIND 4.1 | *Rohou and Grigorieff, 2015* | http://grigori efflab.janelia.org/ctf | |
| Software, algorithm | Relion v 2.1 and 3.0 | *Kimanius et al., 2016* | https:/ /www2.mrc -lmb.cam. ac.uk/relion/ | |
| Software, algorithm | Phenix 1.14 | *Adams et al., 2010* | http://phenix-online.org/ | |
| Software, algorithm | Coot 0.8.9.1 | *Emsley and Cowtan, 2004* | https://www2.mrc-lmb.cam.ac.uk/ personal/pemsley/coot/ | |
| Software, algorithm | Pymol 2.0 | Schrodinger LLC | https://pymol.org/2/ | |
| Software, algorithm | Chimera 1.12 | *Pettersen et al., 2004* | https://www.cgl .ucsf.edu/chimera/ | |
| Software, algorithm | ChimeraX 0.6 | *Goddard et al., 2018* | https://www.rbvi. ucsf.edu/chimerax/ | |
| Other | Whatman Nuclepore Track-Etched Membranes diam. 19 mm, pore size 0.4 μm, polycarbonate | Millipore Sigma | Cat#WHA800282 | |
| Other | HPL6 | Maximator | | |
| Other | Fluoromax 4 | Horiba | | |
| Other | 300 mesh Au 1.2 /1.3 cryo-EM grids | Quantifoil | Cat#N1-C14nAu30-01 | |

## Strains

Wild type *S. cerevisiae* FGY217 were grown either on YPD agar or in YPD liquid media supplemented with 2% glucose at 30°C. After the transformation with respective plasmids, the cells were grown on yeast synthetic drop-out media without uracil with 2% glucose at 30°C. For protein expression, the cells were transferred into a selective media containing 0.1% glucose.

## Construct preparation

The sequence encoding nhTMEM16 was cloned into a modified FX-cloning compatible (*Geertsma and Dutzler, 2011*) pYES2 vector as a C-terminal fusion to streptavidin-binding peptide

(SBP) and Myc tags that were followed by an HRV 3C cleavage site. The mutations were introduced using a QuickChange method (*Zheng et al., 2004*).

## Protein expression and purification

All buffers were prepared using $Ca^{2+}$-free water for molecular biology (Millipore). Plasmids carrying the WT or mutant genes were transformed into the *S. cerevisiae* FGY217 strain as described (*Gietz and Schiestl, 2007*) using lithium acetate/single-stranded carrier DNA/polyethylene glycol method. The cells carrying the plasmid were grown at 30°C in a yeast synthetic dropout media until the culture reached the $OD_{600}$ of 0.8. Afterwards, the protein expression was initialized by adding 2% galactose and the temperature was decreased to 25°C. The protein was expressed for 40 hr after induction. Cells were harvested at 7,200 *g* for 10 min and resuspended in buffer A (50 mM HEPES pH 7.6, 150 mM NaCl, 10 mM EGTA) containing 1 mM $MgCl_2$, DNAse and protease inhibitor cocktail tablets and lysed using a high pressure cell lyser HPL6 at 40 KPsi. Cell debris was removed by centrifugation at 8,000 *g* for 30 min. Membranes were subsequently harvested by centrifugation with a 45 TI rotor (Beckmann) at 200,000 *g* for 1.5 hr, resuspended in buffer A containing 5% glycerol and flash-frozen in liquid $N_2$ and stored at −80°C until further use. During purification, all steps were carried out at 4°C or on ice. For purification, different amount of membrane vesicles were used depending on the purpose of the experiment: 30 g of membranes were used for sample preparation for cryo EM in detergent and in nanodiscs each, and 10 g of membranes for functional assays. Membranes were solubilized in buffer A containing 2% *n*-dodecyl-β-d-maltopyranoside (DDM) and 5% glycerol for 1.5 hr. The insoluble fraction was removed by ultracentrifugation at 200,000 *g* for 30 min. The supernatant was applied to 4 ml of streptavidin Ultralink resin for 1.5 hr, incubated under gentle agitation. The resin containing bound protein was subsequently washed with 50 column volumes (CV) of buffer B (10 mM HEPES pH 7.6, 150 mM NaCl, 5 mM EGTA, 0.03% DDM, 5% glycerol).

For sample preparation for cryo-EM in detergent, the purification tag was cleaved on the resin with 2.4 mg of HRV 3C protease in buffer B containing 20 µg/ml of yeast polar lipids for 2 hr. The flow-through was collected, concentrated using 100 kDa cut-off centrifugal filter units at 700 *g* and injected onto a Superdex 200 size-exclusion column equilibrated in buffer C (5 mM HEPES pH 7.6, 150 mM NaCl, 2 mM EGTA, 0.03% DDM). Main peak fractions were pooled and concentrated to 3.3 mg/ml as described above.

For sample preparation in lipid nanodiscs, membrane scaffold protein (MSP) 2N2 was expressed and purified as described (*Ritchie et al., 2009*), except that the polyhistidine-tag was not removed. Chloroform-solubilized lipids (POPC:POPG at a molar ratio of 7:3) were pooled, dried under a nitrogen stream and washed twice with diethyl ether. The resulting lipid film was dried in a desiccator overnight, and solubilized in 30 mM DDM at a final lipid concentration of 10 mM. nhTMEM16 was purified as described above using the same amount of streptavidin resin, except that the uncleaved fusion construct was eluted from the beads with buffer B containing 3 mM biotin. Protein-containing fractions were pooled and concentrated as described above. Biotin was removed from the sample via gel filtration on a Superdex 200 column. Purified protein was assembled into 2N2 nanodiscs at three different molar ratios of protein:lipids:MSP (*i.e.* 2:725:10, 2:775:10, and 2:1100:10) as described (*Ritchie et al., 2009*). This procedure resulted in a five-fold excess of empty nanodiscs, which prevented multiple incorporations of target proteins into a single nanodisc. nhTMEM16 was mixed with detergent-solubilized lipids and incubated for 30 min on ice. Subsequently, purified 2N2 was added to the sample, and the mixture was incubated for an additional 30 min. Detergent was removed by incubating the sample overnight with SM-2 biobeads (200 mg of beads/ml of the reaction) under constant rotation. From this point on, detergent was excluded from all buffers. To separate the protein-containing from empty nanodiscs, the sample was incubated with 1 ml of streptavidin resin for 1.5 hr. The resin was washed with 10 CV of buffer B and assembled nanodiscs containing nhTMEM16 were eluted in buffer B containing 3 mM biotin. The purification tag was removed by incubation with 0.8 mg of HRV 3C protease for 2 hr. Cleaved samples were concentrated at 500 *g* using concentrators (Amicon) with a molecular weight cut-off of 100 kDa and injected onto a Superose 6 column equilibrated in buffer C. Analogously to the detergent sample, the main peak fractions were concentrated to around 2 mg/ml.

For reconstitution into proteoliposomes, nhTMEM16 was purified as described for detergent samples used for cryo-EM with minor differences: lysate was incubated with 3 ml of streptactin resin and eluted in buffer B containing 5 mM of d-desthiobiotin and the purification tag was not removed.

## Reconstitution into the liposomes and scrambling assay

Functional reconstitution was carried out essentially as described (*Malvezzi et al., 2013*). Lipids for the reconstitution (*E. coli* polar extract and egg-PC at a ratio of 3:1 (w/w) supplemented with 0.5% (w/w) 18:1-06:0 NBD-PE) were pooled and dried as described above. The lipid film was dissolved in assay buffer A (20 mM HEPES pH 7.5, 300 mM KCl, 2 mM EGTA) containing 35 mM CHAPS at a final lipid concentration of 20 mg/ml, aliquoted, flash-frozen and stored until further use. Soy bean lipids extract with 20% cholesterol were prepared as described in *Alvadia et al., 2019*. On the day of reconstitution, lipids were diluted to 4 mg/ml in assay buffer A and purified protein was added to the lipid mixture at a lipid to protein ratio of 300:1 (w/w). As a control, an equivalent volume of the gel filtration buffer was added to aliquots from the same batch of lipids and treated the same way as the proteoliposomes sample, resulting in the assembly of empty liposomes. Soy bean lipid mixture was destabilized with Triton X-100 as described in *Alvadia et al., 2019*. The mixture was incubated for 15 min at room temperature (RT). Biobeads (15 mg of beads/mg of lipids) were added 4 times (after 15 min, 30 min, 1 hr and 12 hr). After the second addition of biobeads, the sample was transferred to 4°C. The proteoliposomes were harvested by centrifugation (150,000 $g$, 30 min, 4°C), resuspended in assay buffer A containing desired amounts of free $Ca^{2+}$ (calculated with the WEB-MAXC calculator) at a final concentration of 10 mg/ml in aliquots of 250 μl, flash-frozen and stored at −80°C until further use. To characterize the protein activity in the lipid mix used for the preparation of nanodisc samples, nhTMEM16 was reconstituted into liposomes composed of 7 POPC:3 POPG (mol/mol) with 0.5% 18:1-06:0 NBD-PE. For conditions mimicking a eukaryotic membrane, the protein was reconstituted in soybean polar lipids extract with 20% cholesterol (mol/mol) and 0.5% 18:1-06:0 NBD-PE. Each mutant was purified and reconstituted independently three times on three different days (defined as distinct biological replicate). To correct for differences in the reconstitution efficiency, WT was included alongside with the mutants into each experiment, and reconstituted into the same batch of lipids on the same day. Reconstitution of the WT into nanodisc lipids and soy bean lipids was performed once.

On the day of the measurement, 250 μl aliquots of liposomes were frozen and thawed 3 times and extruded 21 times through a 400 nm polycarbonate membrane using LipoFast extruder (Avestin). Scrambling data were recorded on a Horiba Fluoromax spectrometer. For the measurement, liposomes were diluted to 0.2 mg/ml in assay buffer B (80 mM HEPES pH 7.5, 300 mM KCl, 2 mM EGTA) containing corresponding amounts of free $Ca^{2+}$. Sodium dithionite was added after 60 s to a final concentration of 30 mM and the fluorescence decay was recorded for additional 340 s. Traces with deviations in fluorescence values of more than $0.03 \times 10^6$ counts per second within the first 60 s of the measurement or displaying uncharacteristic spikes in fluorescence decay after addition of dithionite were discarded. Each sample was measured three times (technical replicate). Data were normalized as F/Fmax. Under ideal conditions, all liposomes should be unilammelar and the lipid composition of the two leaflets of the bilayer should be symmetric. Such scenario would result in a 50% fluorescence decrease upon addition of dithionite to the outside of empty liposomes and a full decay of the fluorescence in liposomes containing an active scramblase, which facilitates lipid flip-flop. In our experiments, we observed fluorescence decays to plateau values ranging between 44–60% in control liposomes and 10–20% in preparations of proteoliposomes containing nhTMEM16 constructs due to variabilities in liposome preparation. The incomplete decay is likely a consequence of multilamellar liposomes and due to a fraction of liposomes not containing an active scramblase in proteoliposome preparations. To account for differences in reconstitution efficiency, all protein constructs that are directly compared to each other in our study and that are shown on the same panel were reconstituted with the same batch of lipids on the same day and the analysis is restricted to a phenotypical comparison of the decay kinetics. Reconstitution efficiency of mutants as compared to wild type nhTMEM16 was estimated by Western blotting using a semi-dry transfer protocol. Protein was transferred onto a PVDF membrane (Immobilon-P) and detected using mouse anti-c-Myc antibody (Millipore Sigma) as a primary (1:5000 dilution) and goat anti-mouse coupled to a peroxidase (Jackson Immunoresearch) as a secondary antibody (1:10,000 dilution).

## Cryo-electron microscopy sample preparation and imaging

2.5 µl of freshly purified protein at a concentration of 3.3 mg/ml when solubilized in DDM and at about 2 mg/ml when reconstituted in nanodiscs were applied on holey-carbon cryo-EM grids (Quantifoil Au R1.2/1.3, 200, 300 and 400 mesh), which were prior glow-discharged at 5 mA for 30 s. For datasets of $Ca^{2+}$-bound protein, samples (containing 2 mM EGTA) were supplemented with 2.3 mM $CaCl_2$ 30 min before freezing, resulting in a free calcium concentration of 300 µM. The pH change in response to the addition of $Ca^{2+}$ was monitored and found negligible. Grids were blotted for 2–5 s in a Vitrobot (Mark IV, Thermo Fisher) at 10–15°C and 100% humidity, plunge-frozen in liquid ethane and stored in liquid nitrogen until further use. Cryo-EM data were collected on a 200 keV Talos Arctica microscope (Thermo Fisher) at the University of Groningen using a post-column energy filter (Gatan) in zero-loss mode, a 20 eV slit, a 100 µm objective aperture, in an automated fashion using EPU software (Thermo Fisher) on a K2 summit detector (Gatan) in counting mode. Cryo-EM images were acquired at a pixel size of 1.012 Å (calibrated magnification of 49,407x), a defocus range from –0.5 to –2 µm, an exposure time of 9 s with a sub-frame exposure time of 150 ms (60 frames), and a total electron exposure on the specimen of about 52 electrons per $Å^2$. The best regions on the grid were screened and selected with an in-house written script to calculate the ice thickness and data quality was monitored on-the-fly using the software FOCUS (*Biyani et al., 2017*).

## Image processing

For the detergent dataset collected in presence of $Ca^{2+}$, the 2521 dose-fractionated cryo-EM images recorded (final pixel size 1.012 Å) were subjected to motion-correction and dose-weighting of frames by MotionCor2 (*Zheng et al., 2017*). The CTF parameters were estimated on the movie frames by ctffind4.1 (*Rohou and Grigorieff, 2015*). Images showing contamination, a defocus above –0.5 or below –2 µm, or a poor CTF estimation were discarded. The resulting 2023 images were used for further analysis with the software package RELION2.1 (*Kimanius et al., 2016*). Around 4000 particles were initially manually picked from a subset of the dataset and classified to create a reference for autopicking. The final round of autopicking on the whole dataset yielded 251,693 particles, which were extracted with a box size of 220 pixels and initial classification steps were performed with two-fold binned data. False positives were removed in the first round of 2D classification. Remaining particles were subjected to several rounds of 2D classification, resulting in 174,901 particles that were further sorted in several rounds of 3D classification. A map created from the X-ray structure of nhTMEM16 (PDBID: 4WIS) was low-pass filtered to 50 Å and used as initial reference for the first round of 3D classification. The resulting best output class was used as new reference in subsequent jobs in an iterative way. The best 3D classes, comprising 128,648 particles, were subjected to auto-refinement, yielding a map with a resolution of 4.3 Å. In the last refinement iteration, a mask excluding the micelle was used and the refinement was continued until convergence (focused refinement), which improved the resolution to 4.0 Å. The final map was masked and sharpened during post-processing resulting in a resolution of 3.9 Å. Finally, the newly available algorithms for CTF refinement and Bayesian polishing implemented in Relion3.0, were applied to further improve the resolution (*Zivanov et al., 2018*). A final round of 3D classification was performed, resulting in 120,086 particles that were subjected to refinement, providing a mask generated from the final PDB model in the last iteration. The final map at 3.6 Å resolution was sharpened using an isotropic b-factor of −126 $Å^2$. While a C1 symmetry was tested and applied throughout image processing workflow, no indication of an asymmetry was identified. Therefore, a C2 symmetry was imposed during the final 3D classification and auto-refinement. Local resolution was estimated by RELION. All resolutions were estimated using the 0.143 cut-off criterion (*Rosenthal and Henderson, 2003*) with gold-standard Fourier shell correlation (FSC) between two independently refined half-maps (*Scheres and Chen, 2012*). During post-processing, the approach of high-resolution noise substitution was used to correct for convolution effects of real-space masking on the FSC curve (*Chen et al., 2013*). The directional resolution anisotropy of density maps was quantitatively evaluated using the 3DFSC web interface (https://3dfsc.salk.edu) (*Tan et al., 2017*; *Table 1*).

For the other datasets, a similar workflow for image processing was applied (*Tables 1* and *2*). In case of the detergent dataset collected in absence of $Ca^{2+}$, a total of 570,203 particles were extracted with a box size of 240 pixels from 2947 images. Several rounds of 2D and 3D classification resulted in a final number of 295,219 particles, which yielded a 3.9 Å map after refinement and post-

processing. CTF refinement and Bayesian polishing were applied, followed by a second round of CTF refinement. One last round of 3D classification was performed, leading to a final pool of 238,070 particles which, after refinement and post processing, resulted in a map at 3.7 Å resolution. For structure determination of nhTMEM16 in lipid nanodiscs, small datasets (around 1000 images) recorded from samples reconstituted at different lipid to protein ratios (LPR) in absence of $Ca^{2+}$ during nanodisc assembly, identified the lowest LPR as optimal condition for large-scale data collection. The dataset in nanodisc in absence of $Ca^{2+}$ resulted in 1,379,187 auto-picked particles from 6465 images extracted with a box size of 240 pixels, which were reduced to 150,421 particles after several rounds of 2D and 3D classification. CTF refinement and Bayesian polishing followed by a final round of 3D classification was performed, resulting in a selection of 133,961 particles. The final refinement and masking resulted in a 3.8 Å resolution map.

Finally, for the dataset in nanodiscs in the presence of $Ca^{2+}$, 2,440,110 particles were picked from 9426 images and extracted with a box size of 240 pixels. Since the dataset displayed high conformational heterogeneity, C1 symmetry was applied throughout the processing workflow. After several rounds of 2D and 3D classification, the best classes (319,530 particles) were combined and refined. The resulting map was used for CTF refinement in Relion3. Refined particles were subjected to 3D refinement and the obtained map was used to generate a mask for nanodisc signal subtraction. The resulting subtracted particles were used for another round of 3D classification, yielding a subset of 279,609 particles. It was evident from the processing results that the core of the protein was stable and displayed a single conformation, whereas α-helices 3 and 4 appeared to be mobile. To separate distinct conformational states, all the particles were aligned using 3D refinement and subsequently subjected to a 3D classification without alignment, using the angles obtained from the preceding refinement job. In one of the resulting classes, the density corresponding to helices 3, 4 and 6 of one of the monomers was not defined, and these particles (11,25%) were thus excluded from further processing. Of the remaining particles, 36,5% are heterogeneous in the region corresponding to α-helices 3 and 4, 14,89% represent nhTMEM16 in a closed conformation of the subunit cavity, 11,91% in the intermediate, and two classes (consisting of 15,64% and 9,81% particles respectively) display an open conformation. Each class was subsequently initially refined applying C1 symmetry. Since no difference between the conformations of individual subunits in the dimeric protein was detected, the refinement was repeated with C2 symmetry imposed. The two open classes were merged and refined together. Similar to other datasets, masks generated from PDB models were applied in the last iteration of the refinement. After refinement and post-processing, the resulting resolutions were 3.57 Å for the open class, 3.57 Å for the closed class and 3.68 Å for the intermediate class. We also attempted to classify the heterogeneous class further, but could not improve the separation of the particles. To generate the map of the open conformation containing the nanodisc signal, we reverted the particles from the open class to their original non-subtracted equivalent, and refined them applying C2 symmetry. This map was used to prepare *Figure 3—figure supplement 3*, *Figure 6* and *Video 2*.

## Model building refinement and validation

For the detergent dataset in the presence of $Ca^{2+}$, the X-ray structure of nhTMEM16 (PDBID: 4WIS) was used as a template. The model was manually edited in Coot (*Emsley and Cowtan, 2004*) prior to real-space refinement in Phenix imposing symmetry restraints between subunits throughout refinement (*Adams et al., 2010*). The higher quality of the cryo-EM density of the cytoplasmic domains compared to the X-ray density allowed for the identification of an incorrect register at the N-terminus of the X-ray structure (PDBID: 4WIS) between residues 20–61. The region comprising residues 15–61 was thus rebuilt. Another change compared to the X-ray structure concerns the remodeled conformation of Lys 598. In several cases the cryo-EM density allowed the interpretation of residues that were not defined in the X-ray structure. These concern residues 130–140 at a loop connecting the cytoplasmic and the transmembrane domain, an extra helical turn at the cytoplasmic side of α6 and residues 476–482 at the loop connecting α6 to α6'. Since residues 690–698 appeared to be helical in the cryo-EM map, the region was rebuilt and it was possible to add the missing residues 688–691. Several regions of the cryo-EM map displayed weaker density compared to the X-ray structure, and hence the residues 416–418, 653–656 and 660–663 had to be removed from the model. This corrected nhTMEM16 structure (PDBID: 6QM5) was used for refinement of the nhTMEM16 structure in DDM in the absence of $Ca^{2+}$, and for model building and refinement of the

$Ca^{2+}$-bound open and $Ca^{2+}$-bound closed structures in nanodiscs. All models were edited manually in Coot, followed by real space refinement in Phenix. The $Ca^{2+}$-free structure in DDM (PDBID: 6QM9) and the $Ca^{2+}$-bound open structure in nanodisc (PDBID: 6QM9) closely resemble the $Ca^{2+}$-bound structure in DDM, displaying only minor differences. Since the density corresponding to $\alpha6$ is better resolved in the $Ca^{2+}$-bound open structure in nanodisc, the helix was extended to Ser 471. The density corresponding to $C\alpha2$ of the C-terminus (residues 664–684) was comparably weak in all three open structures and the helix was thus fitted as a rigid body. For the $Ca^{2+}$-bound closed conformation (PDBID: 6QMB), major changes were introduced for $\alpha3$, $\alpha4$ and to a lesser extent for $\alpha6$. The densities of the loop connecting $\alpha5'$ and $\alpha6$ and of the cytosolic region of $\alpha6$ were generally weaker and residues 417–424 and 467–471 were thus not interpreted in this structure. Since the $Ca^{2+}$-bound closed map in nanodisc displays the highest resolution and quality in the region encompassing $\alpha3$ and $\alpha4$, the resulting model was used as a template for the interpretation of the $Ca^{2+}$-bound intermediate (PDBID: 6QMA) and $Ca^{2+}$-free (PDBID: 6QM4) structures in nanodisc. In case of the $Ca^{2+}$-bound intermediate structure, introduced changes are mostly confined to $\alpha4$, where the extracellular part of the $\alpha$-helix undergoes a conformational transition towards opening of the subunit cavity. For the $Ca^{2+}$-free structure in nanodisc, the residues 461–467 of $\alpha6$ were not modelled due to its high mobility. Figures were prepared using Pymol (The PyMOL Molecular Graphics System, Version 2.0 Schrödinger, LLC), Chimera (*Pettersen et al., 2004*) and ChimeraX (*Goddard et al., 2018*).

## Data availability

The three-dimensional cryo-EM density maps of calcium-free nhTMEM16 in detergent and nanodiscs have been deposited in the Electron Microscopy Data Bank under accession numbers EMD-4589 and EMD-4587, respectively. The maps of calcium-bound samples in detergent and calcium-bound open, calcium-bound intermediate and calcium-bound closed in nanodiscs were deposited under accession numbers EMD-4588, EMD-4592, EMD-4593 and EMD-4594, respectively. The deposition includes the cryo-EM maps, both half-maps, the unmasked and unsharpened refined maps and the mask used for final FSC calculation. Coordinates of all models have been deposited in the Protein Data Bank. The accession numbers for calcium-bound and calcium-free models in detergents are 6QM5 and 6QM6, respectively. The accession numbers for calcium-bound open, calcium-bound intermediate, calcium-bound closed and calcium-free models in nanodiscs are 6QM9, 6QMA, 6QMB and 6QM4, respectively.

## Acknowledgments

We thank S Klauser, S Rast and M Punter for their help in establishing the computer infrastructure, H Stahlberg and K Goldie at C-Cina of the University of Basel for access to cryo-electron microscopes at an initial stage of the project. S Weidner and the Workshop of Biochemistry department (UZH) are acknowledged for developing the high-pressure cell lyser. D Deneka is acknowledged for his advice on nanodisc reconstitution. C Alvadia and N Lim are acknowledged for helpful discussions and their input during manuscript preparation. All members of the Dutzler and Paulino labs are acknowledged for help at various stages of the project.

## Additional information

### Funding

| Funder | Grant reference number | Author |
|---|---|---|
| FP7 Ideas: European Research Council | 339116 Anobest | Raimund Dutzler |
| Nederlandse Organisatie voor Wetenschappelijk Onderzoek | 740.018.016 | Cristina Paulino |

The funders had no role in study design, data collection and interpretation, or the decision to submit the work for publication.

## Author contributions

Valeria Kalienkova, Formal analysis, Validation, Investigation, Visualization, Writing—original draft, Writing—review and editing, Purified proteins for cryo-EM and functional characterization, Reconstituted protein into nanodiscs and liposomes and carried out lipid transport experiments, Prepared the samples for cryo-EM, Collected cryo-EM data, Carried out image processing, model building and refinement, Jointly planned experiments, analyzed the data and wrote the manuscript; Vanessa Clerico Mosina, Formal analysis, Validation, Investigation, Visualization, Writing—original draft, Writing—review and editing, Collected cryo-EM data, Carried out image processing, model building and refinement, Jointly planned experiments, analyzed the data and wrote the manuscript; Laura Bryner, Investigation, Purified proteins for cryo-EM and functional characterization; Gert T Oostergetel, Formal analysis, Collected cryo-EM data; Raimund Dutzler, Conceptualization, Resources, Formal analysis, Supervision, Funding acquisition, Validation, Visualization, Writing—original draft, Project administration, Writing—review and editing, Jointly planned experiments, Analyzed the data and wrote the manuscript; Cristina Paulino, Conceptualization; Resources; Formal analysis; Supervision; Funding acquisition; Validation; Investigation; Visualization; Writing—original draft; Project administration; Writing—reviewand editing; prepared the samples for cryo-EM; collected cryo-EM data; carried out image processing, model building and refinement; jointly planned experiments; analyzed the data and wrote the manuscript

## Author ORCIDs

Valeria Kalienkova (iD) https://orcid.org/0000-0002-4143-6172
Vanessa Clerico Mosina (iD) https://orcid.org/0000-0001-8013-0144
Gert T Oostergetel (iD) https://orcid.org/0000-0001-6816-136X
Raimund Dutzler (iD) https://orcid.org/0000-0002-2193-6129
Cristina Paulino (iD) https://orcid.org/0000-0001-7017-109X

## Decision letter and Author response

Decision letter https://doi.org/10.7554/eLife.44364.053
Author response https://doi.org/10.7554/eLife.44364.054

## Additional files

### Supplementary files

• Transparent reporting form
DOI: https://doi.org/10.7554/eLife.44364.027

### Data availability

The three-dimensional cryo-EM density maps of calcium-free nhTMEM16 in detergent and nanodiscs have been deposited in the Electron Microscopy Data Bank under accession numbers EMD-4589 and EMD-4587, respectively. The maps of calcium-bound samples in detergent and calcium-bound open, calcium-bound intermediate and calcium-bound closed in nanodiscs were deposited under accession numbers EMD-4588, EMD-4592, EMD-4593 and EMD-4594, respectively. The deposition includes the cryo-EM maps, both half-maps, the unmasked and unsharpened refined maps and the mask used for final FSC calculation. Coordinates of all models have been deposited in the Protein Data Bank. The accession numbers for calcium-bound and calcium-free models in detergents are 6QM5 and 6QM6, respectively. The accession numbers for calcium-bound open, calcium-bound intermediate, calcium-bound closed and calcium-free models in nanodiscs are 6QM9, 6QMA, 6QMB and 6QM4, respectively.

The following datasets were generated:

| Author(s) | Year | Dataset title | Dataset URL | Database and Identifier |
|---|---|---|---|---|
| Kalienkova V, Clerico Mosina V, Laura Bryner, Gert T Oostergetel, Raimund Dutzler, Cris- | 2019 | Cryo-EM structure of calcium-free nhTMEM16 lipid scramblase in nanodisc | https://www.ebi.ac.uk/pdbe/entry/emdb/EMD-4587 | Electron Microscopy Data Bank, EMD-4587 |

tina Paulino

| | | | | |
|---|---|---|---|---|
| Kalienkova V, Clerico Mosina V, Bryner L, Ooster-getel GT, Dutzler R | 2019 | Cryo-EM structure of calcium-bound nhTMEM16 lipid scramblase in DDM | https://www.rcsb.org/structure/6QM5 | Protein Data Bank, 6QM5 |
| Kalienkova V, Clerico Mosina V, Bryner L, Ooster-getel GT, Dutzler R, Paulino C | 2019 | Cryo-EM structure of calcium-free nhTMEM16 lipid scramblase in DDM | https://www.rcsb.org/structure/6QM6 | Protein Data Bank, 6QM6 |
| Kalienkova V, Clerico Mosina V, Bryner L, Ooster-getel GT, Dutzler R, Paulino C | 2019 | Cryo-EM structure of calcium-bound nhTMEM16 lipid scramblase in nanodisc (open state) | https://www.rcsb.org/structure/6QM9 | Protein Data Bank, 6QM9 |
| Kalienkova V, Clerico Mosina V, Bryner L, Ooster-getel GT, Dutzler R, Paulino C | 2019 | Cryo-EM structure of calcium-bound nhTMEM16 lipid scramblase in nanodisc (intermediate state) | https://www.rcsb.org/structure/6QMA | Protein Data Bank, 6QMA |
| Kalienkova V, Clerico Mosina V, Bryner L, Ooster-getel GT, Dutzler R, Paulino C | 2019 | Cryo-EM structure of calcium-bound nhTMEM16 lipid scramblase in nanodisc (closed state) | https://www.rcsb.org/structure/6QMB | Protein Data Bank, 6QMB |
| Kalienkova V, Clerico Mosina V, Bryner L, Ooster-getel GT, Dutzler R, Paulino C | 2019 | Cryo-EM structure of calcium-free nhTMEM16 lipid scramblase in nanodisc | https://www.rcsb.org/structure/6QM4 | Protein Data Bank, 6QM4 |
| Kalienkova V, Clerico Mosina V, Bryner L, Ooster-getel GT, Dutzler R, Paulino C | 2019 | Cryo-EM structure of calcium-bound nhTMEM16 lipid scramblase in DDM | https://www.ebi.ac.uk/pdbe/entry/emdb/EMD-4588 | Electron Microscopy Data Bank, EMD-4588 |
| Kalienkova V, Clerico Mosina V, Bryner L, Ooster-getel GT, Dutzler R, Paulino C | 2019 | Cryo-EM structure of calcium-free nhTMEM16 lipid scramblase in DDM | https://www.ebi.ac.uk/pdbe/entry/emdb/EMD-4589 | Electron Microscopy Data Bank, EMD-4589 |
| Kalienkova V, Clerico Mosina V, Bryner L, Ooster-getel GT, Dutzler R, Paulino C | 2019 | Cryo-EM structure of calcium-bound nhTMEM16 lipid scramblase in nanodisc (open state) | https://www.ebi.ac.uk/pdbe/entry/emdb/EMD-4592 | Electron Microscopy Data Bank, EMD-4592 |
| Kalienkova V, Clerico Mosina V, Bryner L, Ooster-getel GT, Dutzler R, Paulino C | 2019 | Cryo-EM structure of calcium-bound nhTMEM16 lipid scramblase in nanodisc (intermediate state) | https://www.ebi.ac.uk/pdbe/entry/emdb/EMD-4593 | Electron Microscopy Data Bank, EMD-4593 |
| Kalienkova V, Clerico Mosina V, Bryner L, Ooster-getel GT, Dutzler R, Paulino C | 2019 | Cryo-EM structure of calcium-bound nhTMEM16 lipid scramblase in nanodisc (closed state) | https://www.ebi.ac.uk/pdbe/entry/emdb/EMD-4594 | Electron Microscopy Data Bank, EMD-4594 |

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
