## [Decision Letter]

Thank you for submitting your article "Stepwise activation mechanism of the scramblase nhTMEM16 revealed by cryo-EM" for consideration by *eLife*. Your article has been reviewed by three peer reviewers, and the evaluation has been overseen by Kenton Swartz as the Reviewing Editor and Richard Aldrich as the Senior Editor. The following individuals involved in review of your submission have agreed to reveal their identity: Angela Ballesteros (Reviewer #1); Joel Meyerson (Reviewer #2); Stephen Barstow Long (Reviewer #3).

The reviewers have discussed the reviews with one another and the Reviewing Editor has drafted this decision to help you prepare a revised submission.

Summary:

The TMEM16 family of proteins function either as calcium-activated chloride channels (CaCC) or as calcium-activated scramblases that transport lipids and ions across the plasma membrane. Structural information of the calcium-free and calcium-bound conformation of the CaCC TMEM16A has helped us to understand the open and close mechanism of a CaCC TMEM16. Likewise, the structure of the fungal nhTMEM16 in its calcium-bound state revealed the presence of a large cavity built at the protein-membrane interface that enlightened the function of this protein as a lipid scramblase. However, the conformational changes that regulate the activation of the scramblase function in TMEM16 are still poorly understood. In this manuscript, Valeria Kalienkova and colleagues throw structural insight on the conformational changes happening during the activation of nhTMEM16, which could be potentially extrapolated to other scramblases. The Cryo-EM structure of nhTMEM16 in nanodisc without calcium reveals a closed cavity similar to those previously reported for TMEM16A, TMEM16F, and afTMEM16 in their closed conformations. Interestingly, the structure was highly variable in the presence of calcium, presenting several classes along a spectrum of conformations between open and closed ones. Therefore, the Cryo-EM structures of nhTMEM16 in nanodisc presented in this manuscript reveal a total of 5 different states of the protein that could explain the conformational changes leading to the opening of the cavity and activation of lipid scrambling. The structural and biochemical data pinpoint several residues involved in the structural rearrangements during the activation of lipid scrambling and suggest an additional control mechanism that could potentially limit the opening of the scrambling cavity. The manuscript is well written, the structural work is expertly done, and overall the study is appropriate for publication in *eLife*. The following are suggestions for improving the manuscript in revision.

Requested revisions:

1) In the control sample of Figure 1——figure supplement 1B, the dithionite does not bleach the 50% of the fluorescence corresponding to the NBD-labeled lipids on the external leaflet of the soybean liposomes. Please explain this phenomenon. In addition, please describe in the figure legend what was used as a control in these experiments (black line in panels A-C).

2) The authors mentioned that: "nhTMEM16 does not contain a flexible glycine residue at the TM6 hinge, but the movement at the TM6 upon calcium release occurs at a similar region (Figure 2C-E)". However, the dramatic movement of the intracellular side of the TM6 observed in the structure of TMEM16A in the absence of calcium is hard to appreciate or missing in the figures of the nhTMEM16 structures. A superimposition of the nhTMEM16 calcium-bound open and calcium-free closed structures in nanodisc indicating the position of the glycine hinge, or in superposition with the calcium-free TMEM16A structure (PDB ID: 5OYB), would help to visualize the conformational change in TM6. In addition, since the authors mentioned in the Discussion that the conformational change of the TM6 in nhTMEM16 is closer to the one observed for TMEM16F (Discussion, first paragraph), it would be good to include a superposition of the structures of nhTMEM16 and TMEM16K to visualize and compare the conformational change of TM6

3) We felt that the figures should be modified to make it easier for the general reader to readily comprehend what the authors wish to show. It can be disorienting when a single nhTMEM16 subunit is presented without showing its orientation with respect to the second subunit, or when sub-regions are shown without an inset box to show the context in the complete structure. This problem is exacerbated when relevant helices are not labeled. This is especially evident in the following panels (below). These issues could mostly be rectified by perhaps dedicating a figure (perhaps the first in the paper) to delineating all helices, key views, interfaces, and binding sites in the structure. This would then be a helpful visual for the reader to reference throughout. These panels were particularly challenging: Figure 1: B, C, E, F; Figure 1—figure supplement 4: B, D; Figure 2: B-E; Figure 3: A (right), B (right), C-G; Figure 3—figure supplement 1: C; Figure 3—figure supplement 2: A-E; Figure 4: A-E; Figure 5: A-D; Figure 7: A.

4) The distortion of the lipid surroundings in the structures of nhTMEM16 (detergent and nanodisc) in the presence of calcium is shown in Figure 6. However, it would be more informative to include in this figure the membrane distortion observed in the structure of nhTMEM16 in nanodisc in the absence of calcium. The addition of these data would help to appreciate the lipid distortion independently of the presence of calcium (subsection “Protein-induced distortion of the lipid environment”). Panels B and C show the same structure but what is their relative orientation? A rotation arrow would clarify this. Consider labeling the subunit cavity.

5) The authors indicate that the "advantage of a C1 symmetry was tested" but can they clarify what this means? Also, was there evidence of deviation from a hypothetical C2 symmetric structure when data was treated as C1? Some general elaboration on use of C1 for processing would be of value to the reader.

6) While the modification of the surrounding detergent or lipid nanodisc is clear from the data presented, it could be noted in the manuscript more clearly that the nanodisc density represents both lipids and nanodisc protein. In this regard, Introduction, subsection “Protein-induced distortion of the lipid environment” ("lipid molecules"), and Discussion, first paragraph, could be clarified or softened by using verbs like "suggest" rather than "show".

7) The authors should consider depositing the low-pass filtered maps from Figure 6 with the EMDB (or, alternatively, the unsharpened maps). This would be beneficial to people interested in the distortions lipid nanodisc/detergent micelle.

8) Have the authors checked the pH of the sample after adding 2.3 mM calcium chloride? The addition of calcium to EGTA liberates protons and may alter the pH somewhat. Please check the pH and indicate the final pH.

9) Subsection “Structural characterization of nhTMEM16 in detergent”. It might be beneficial to describe the reasoning with regard to inactivation more clearly. Has inactivation been shown? Is the time constant known? Is anything known about the dependence on detergent versus a lipid environment for inactivation?

---

## [Author Response]

Requested revisions:1) In the control sample of Figure 1—figure supplement 1B, the dithionite does not bleach the 50% of the fluorescence corresponding to the NBD-labeled lipids on the external leaflet of the soybean liposomes. Please explain this phenomenon. In addition, please describe in the figure legend what was used as a control in these experiments (black line in panels A-C).

Indeed, in some cases the addition of dithionite does not result in a 50% bleaching, which would correspond to the expected amount of fluorescent lipids on the outer leaflet. This is a known and common problem that originates in the slight differences during reconstitution, i.e. due to the formation of multilamellar vesicles. To account for these differences in reconstitution efficiency and vesicle formation any samples (i.e. control (empty liposomes), WT and mutants) that are directly compared to each other and plotted in the same graph were reconstituted with the same batch of lipids on the same day and the analysis is restricted to a phenotypical comparison of the decay kinetics. We have described this in more detail in the Materials and methods (subsection “Reconstitution into the liposomes and scrambling assay”, last paragraph). We have also specified that we used empty liposomes as a control (see the first paragraph of the aforementioned subsection).

2) The authors mentioned that: "nhTMEM16 does not contain a flexible glycine residue at the TM6 hinge, but the movement at the TM6 upon calcium release occurs at a similar region (Figure 2C-E)". However, the dramatic movement of the intracellular side of the TM6 observed in the structure of TMEM16A in the absence of calcium is hard to appreciate or missing in the figures of the nhTMEM16 structures. A superimposition of the nhTMEM16 calcium-bound open and calcium-free closed structures in nanodisc indicating the position of the glycine hinge, or in superposition with the calcium-free TMEM16A structure (PDB ID: 5OYB), would help to visualize the conformational change in TM6. In addition, since the authors mentioned in the Discussion that the conformational change of the TM6 in nhTMEM16 is closer to the one observed for TMEM16F (Discussion, first paragraph), it would be good to include a superposition of the structures of nhTMEM16 and TMEM16K to visualize and compare the conformational change of TM6

We have included an additional panel in Figure 2 (panel C) showing a superposition of α-helix 6 of nhTMEM16 with TMEM16F and TMEM16A. We have also added a structure-based sequence alignment of selected TMEM16 scramblases with the secondary structure of nhTMEM16 indicated (see Figure 1—figure supplement 4).

3) We felt that the figures should be modified to make it easier for the general reader to readily comprehend what the authors wish to show. It can be disorienting when a single nhTMEM16 subunit is presented without showing its orientation with respect to the second subunit, or when sub-regions are shown without an inset box to show the context in the complete structure. This problem is exacerbated when relevant helices are not labeled. This is especially evident in the following panels (below). These issues could mostly be rectified by perhaps dedicating a figure (perhaps the first in the paper) to delineating all helices, key views, interfaces, and binding sites in the structure. This would then be a helpful visual for the reader to reference throughout. These panels were particularly challenging: Figure 1: B, C, E, F; Figure 1—figure supplement 4: B, D; Figure 2: B-E; Figure 3: A (right), B (right), C-G; Figure 3—figure supplement 1: C; Figure 3—figure supplement 2: A-E; Figure 4: A-E; Figure 5: A-D; Figure 7: A.

Dear reviewers, very valid point we realized upon revision. To better guide the reader through our figures we have added three additional panels in Figure 1, where we label all transmembrane-helices and indicate the orientations used in many subsequent figures. We have also added boxes, detail on rotations and other sorts of guides to the indicated figures and whenever applicable optimized the figure legends. We have also added a structure-based sequence alignment of selected TMEM16 scramblases with the secondary structure of nhTMEM16 indicated (see Figure 1—figure supplement 4) and a superposition of the entire dimer in the open and closed state in Figure 5A.

4) The distortion of the lipid surroundings in the structures of nhTMEM16 (detergent and nanodisc) in the presence of calcium is shown in Figure 6. However, it would be more informative to include in this figure the membrane distortion observed in the structure of nhTMEM16 in nanodisc in the absence of calcium. The addition of these data would help to appreciate the lipid distortion independently of the presence of calcium (subsection “Protein-induced distortion of the lipid environment”). Panels B and C show the same structure but what is their relative orientation? A rotation arrow would clarify this. Consider labeling the subunit cavity.

We have changed the figure accordingly and show now the micelle/nanodisc distortion in all four datasets. Rotation is also indicated. As before Video 2 shows the distortion for all four datasets as well.

5) The authors indicate that the "advantage of a C1 symmetry was tested" but can they clarify what this means? Also, was there evidence of deviation from a hypothetical C2 symmetric structure when data was treated as C1? Some general elaboration on use of C1 for processing would be of value to the reader.

It is good conduct to test a putative existing symmetry by processing the dataset without imposing the same first, to check if the particles remain symmetrical. This approach can also be beneficial in initial processing steps – before masking or subtraction is applied – when working with membrane proteins, which have a less resolved and potential asymmetric/inhomogeneous surrounding, i.e. with large nanodiscs. Further, it was shown for one of the family members (namely TMEM16A) that the individual subunits are activated independently (Lim et al., 2016, Jeng et al., 2016). For these reasons, we applied C1 symmetry during the initial processing. However, since we did not observe obvious differences in conformations of subunits, C2 symmetry was applied at the final stages of data processing. We have elaborated more on this matter in Materials and methods (subsection “Image Processing”, last paragraph).

6) While the modification of the surrounding detergent or lipid nanodisc is clear from the data presented, it could be noted in the manuscript more clearly that the nanodisc density represents both lipids and nanodisc protein. In this regard, Introduction, subsection “Protein-induced distortion of the lipid environment” ("lipid molecules"), and Discussion, first paragraph, could be clarified or softened by using verbs like "suggest" rather than "show".

To clarify the source of the entire density observed for nanodiscs as shown in Figure 6, we have added to the figure legend the following: “The density corresponding to the detergent micelle or the nanodisc, which is composed of lipids surrounded by the 2N2 belt protein, are colored […]”. We would however like to emphasize that the distortion observed in direct vicinity to nhTMEM16, as shown in Figure 6B, E derives from the phospholipid headgroups and not from the MSP belt protein.

7) The authors should consider depositing the low-pass filtered maps from Figure 6 with the EMDB (or, alternatively, the unsharpened maps). This would be beneficial to people interested in the distortions lipid nanodisc/detergent micelle.

To facilitate this interpretation, we have deposited alongside also the unmasked and unsharpened refined maps for four datasets (DDM with and without Ca^2+^, 2N2 without Ca^2+^, 2N2 with Ca^2+^, open state). For the map of the subtracted Ca^2+^-bound open state of nhTMEM16 in nanodiscs we reverted the particles to their original non-subtracted equivalent, and refined them with symmetry C2 (subsections “Image Processing”, last paragraph and “Data availability”).

8) Have the authors checked the pH of the sample after adding 2.3 mM calcium chloride? The addition of calcium to EGTA liberates protons and may alter the pH somewhat. Please check the pH and indicate the final pH.

The pH values were measured before and after addition of Ca^2+^ ions. We have added a respective sentence in material methods as follows: “The pH change in response to the addition Ca^2+^ was monitored and found negligible.”

*9) Subsection “Structural characterization of nhTMEM16 in detergent”. It might be beneficial to describe the reasoning with regard to inactivation more clearly. Has inactivation been shown? Is the time constant known? Is anything known about* the dependence on detergent versus a lipid environment for inactivation?

In light of the compromised ion conduction properties of TMEM16A and TMEM16F after prolonged exposure to high Ca^2+^ concentrations (Ye et al., 2014; Ni et al., 2014; Ye et al., 2018) one could question if the X-ray structure of nhTMEM16, where the protein was crystallized in 3 mM Ca^2+^, might represents an inactive conformation. However, the structural similarity between the X-ray and the cryo-EM structure of nhTMEM16 in detergent, where calcium was added just before freezing and at a lower concentration, refutes such concerns. Together with the fact that we can sample the conformational transition of nhTMEM16 from an open to a closed state in the Ca^2+^-bound dataset in nanodisc emphasizes that the observed open conformation most likely displays an active state of the protein. After revision we however believe that the manuscript does not benefit from this discussion. It would rather confuse the reader, in particular because this effect was never discussed before for nhTMEM16. We have thus deleted the statement.